# Detecting Propaganda Techniques in Code-Switched Social Media Text

**Muhammad Umar Salman, Asif Hanif, Shady Shehata, Preslav Nakov**
Mohamed Bin Zayed University of Artificial Intelligence (MBZUAI)
{umar.salman,asif.hanif,shady.shehata,preslav.nakov}@mbzuai.ac.ae

## Abstract

Propaganda is a form of communication intended to influence the opinions and the mindset of the public to promote a particular agenda. With the rise of social media, propaganda has spread rapidly, leading to the need for automatic propaganda detection systems. Most work on propaganda detection has focused on high-resource languages, such as English, and little effort has been made to detect propaganda for low-resource languages. Yet, it is common to find a mix of multiple languages in social media communication, a phenomenon known as *code-switching*. Code-switching combines different languages within the same text, which poses a challenge for automatic systems. Considering this premise, we propose a novel task of detecting propaganda techniques in code-switched text. To support this task, we create a corpus of 1,030 texts code-switching between English and Roman Urdu, annotated with 20 propaganda techniques at the fragment level. We perform a number of experiments contrasting different experimental setups, and we find that it is important to model the multilinguality directly rather than using translation as well as to use the right fine-tuning strategy. The code and the dataset are publicly available at https://github.com/mbzuai-nlp/propaganda-codeswitched-text

## 1 Introduction

The rise of social media has brought a significant change in the way people consume and share information. With the advent of the Internet and the widespread use of social media platforms, it has become easier for anyone to spread information to a wide audience. Social media platforms in the digital era have become the primary source of news consumption and are used on a daily basis.

Unfortunately, this democratic channel was also used by malicious users for harmful purposes, e.g., to spread disinformation and hate speech. An important tool to make such content believable and viral is the use of propaganda techniques. Propaganda is the dissemination of biased or misleading information in order to manipulate people's beliefs and opinions towards a particular objective. Propaganda has always existed, but social media platforms have made it easier for individuals and organizations to promote their agenda and narratives quickly across large audiences. Propaganda and the spread of misleading claims and opinions negatively affects many people and may cause harm. This was particularly seen during the COVID-19 pandemic when incorrect details about the virus and the effectiveness of vaccines spread on social media, leading to confusion and mistrust. Propaganda is also used as a tool to manipulate people's emotions and behavior, often for political or ideological purposes. This can result in a decline in critical thinking abilities, impairing the capacity to make well-informed decisions grounded in factual information and evidence. Thus, there is a need to detect the use of propaganda and to combat the negative impact it poses as well as to promote a more accurate and truthful exchange of information in social media.

Social media platforms such as Facebook, Twitter, YouTube, and Instagram are used by millions of people around the world, and many of these users are bilingual and even multilingual. Due to the increase of multilingual users and informal interactions on social media, a large number of people have begun to resort to mixing multiple languages on various social media platforms in the form of posts, tweets, comments, and especially chats. This is where the term *code-switching* comes into play. Code-switching primarily refers to switching or alternating between two or more languages in the same context (Sitaram et al., 2019).

---

WARNING: This paper contains examples and words that are offensive in nature.

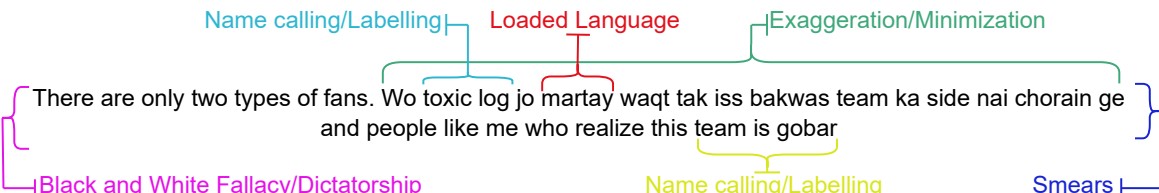

Figure 1: Example of code-switched text annotated at the fragment level. Here, the entire text is labeled as *Smears* and *Black and White Fallacy/Dictatorship* **Translation of the text:** *There are only two types of fans. Those toxic people who till death will not leave this rubbish team's side and people like me who realize this team is dung.*

This switching may occur within the same sentence or between neighboring sentences. This phenomenon of code-switching or mixing multiple languages has now become prevalent on social media platforms among bilingual and multilingual communities. It often compensates for a lack of proper expression. This helps individuals to express their ideas and thoughts more accurately and convincingly and to achieve a better impact on the readers and the listeners (Nilep, 2006; Tay, 1989; Scotton, 1982). In this work, we aim to detect propaganda techniques in code-switched text: we use English and Roman Urdu (i.e., Urdu language written using Latin script) as our high-resource and low-resource languages. Although Urdu is typically written in its native Arabic script, most online users prefer to use Roman Urdu in their online textual communication due to the convenience of typing. Figure 1 shows an example of a code-switched text (English and Roman Urdu), where the different fragments of the text are labeled as propaganda techniques.

Significant research efforts have been dedicated to various code-switched tasks in the field of Natural Language Processing (NLP), such as Language Identification, Named Entity Recognition (NER), POS Tagging, Sentiment Analysis, Question Answering, and Natural Language Inference (NLI) (Khanuja et al., 2020; Jose et al., 2020; Chen et al., 2022; Rizwan et al., 2020). However, there has been limited exploration in the domain of propaganda detection, particularly for low-resource languages. Previous work in this area has focused mainly on high-resource languages such as English, with propaganda detection at the document-level as both a binary-class and a multi-class classification task (Barrón-Cedeño et al., 2019; Rashkin et al., 2017), as well as at the fragment-level for journalistic news articles (Da San Martino et al., 2019). Our research represents the first attempt to address propaganda detection in code-switched low-resource languages on social media.

However, we face several challenges in this task, including a lack of specialized corpora, high-quality and reliable annotations, and fine-tuned models. Our approach aims to overcome these challenges by focusing on propaganda detection on social media platforms. We can summarize the contributions of our work as follows:

1. We formulate the novel NLP task of detecting propaganda techniques in code-switched text.

2. We construct and annotate a new corpus specifically for this task, comprising 1,030 code-switched texts in English and Roman Urdu. These texts are annotated at the fragment level with 20 propaganda techniques.

3. We experiment with various model classes, including monolingual, multilingual, cross-lingual models, and Large Language Models (LLMs), for this task and dataset, and we provide a detailed comparative performance analysis, which can inform future research.

## 2 Related Work

Propaganda originated in the $17^{th}$ century in the form of agendas that were propagated during gatherings and events such as carnivals, fairs, religious festivals, and theatres (Margolin, 1979). From there, propaganda took the form of text and pictures in newspapers and posters. During the American Revolution in the $18^{th}$ century, newspapers and the printing press in various colonies propagated their views to promote patriotism (Cole, 1975). During the First World War, the British government organized a large-scale propaganda campaign, which was claimed by military officials to have played a key role in the defeat of the Germans in 1914 (Yourman, 1939). Later, in the early $20^{th}$ century, after the advent of motion pictures, this medium became a new propaganda tool used to promote military and political agendas.

The Artificial Intelligence (AI) community has primarily focused on textual propaganda. With the widespread accessibility of the Internet in the past two decades, propaganda has proliferated at an enormous scale, compelling research communities to devote efforts towards detecting propaganda in its various forms. Initially, research on propaganda in AI started at the document level. Barrón-Cedeño et al. (2019) performed binary-class propaganda detection at the news article level. Their model assessed the level of propaganda based on the presence of keywords, specific writing style, and readability level. Rashkin et al. (2017) focused on distinguishing between four types of documents in their annotated corpus (TSHP-17): *satire*, *hoax*, *trusted*, and *propaganda*. They took an analytical approach to the language of the news media in the context of political fact-checking and fake news detection.

Da San Martino et al. (2019) took a more fine-grained approach to propaganda detection. To counter the lack of interpretability caused by binary labeling of the entire article, they proposed to detect propaganda techniques in spans/fragments of the text. The annotation in their corpus was at the fragment level, and featured 18 propaganda techniques. This allows for two tasks: *(i)* binary classification (whether propaganda is present in a sentence or not), and *(ii)* multi-label classification (where a span of text in a sentence can be labeled as one or more classes from a total of 18 techniques). The list of the 18 propaganda techniques was selected due to their occurrence in journalistic articles. This task led to the creation of the PTC corpus (Da San Martino et al., 2019), which contains 451 news articles from 48 news outlets.

There were also some related shared tasks. SemEval-2020 Task 11 (Da San Martino et al., 2020) focused on detecting persuasion techniques in news articles. The task involved identifying specific text spans and predicting their corresponding types (14 different techniques). Dimitrov et al. (2021b) extended the scope of fragment-level propaganda identification in SemEval-2021 Task 6: the task includes the detection of propaganda techniques present in both textual and image data. In a WANLP'2022 shared task, Alam et al. (2022) focused on detecting propaganda techniques in Arabic, specifically using Arabic tweets. A follow up WANLP'2023 task, focused on propaganda in tweets and news articles (Hasanain et al., 2023).

Moreover, Piskorski et al. (2023a) (SemEval-2023 Task 3) worked on detecting the category, the framing, and the persuasion techniques in a multilingual setup. This work makes use of news articles in nine different languages during training and evaluates on additional three languages in the testing phase.

(Dimitrov et al., 2021a) focused on multimodal propaganda detection, as a multi-label multimodal task that focuses on propaganda detection in memes. They released a corpus of 950 memes, labeled with 22 propaganda techniques. Yu et al. (2021) developed an interpretable model for propaganda detection. More recently, Baleato Rodríguez et al. (2023) performed multi-task learning with propaganda identification as the main task and metaphor detection as an auxiliary task. Their work makes use of the close relationship between propaganda and metaphors. They leveraged metaphors which in turn improves the model performance, especially while identifying *name calling* and *loaded language*. Piskorski et al. (2023b) presented a new multilingual multi-facet dataset comprising 1,612 news articles used for understanding the news (particularly "fake news"). The dataset includes annotated documents categorized by genre, framing, and rhetoric (persuasion). The annotation of persuasion techniques is conducted at the span level, using a taxonomy that consists of 23 techniques organized into six categories.

## 3 Dataset

### 3.1 Data Collection

We collected data from publicly available sources, including Twitter, Facebook, Instagram, and YouTube. Twitter contributes 60% of the dataset, Facebook 25%, Instagram 10%, and the remaining 5% are from YouTube. Four data collectors were selected based on the quality of their submission of 20 examples of code-switched text. They were then instructed to collect a total of 500 texts each. We ensured diversity among the collectors by including two male and two female collectors. The collectors were also advised to gather texts at different times of the day and from different time periods to increase the diversity of trending topics. After some filtering, we ended up with 1,030 examples. Some examples were removed as they required extensive prior knowledge to understand the annotations, while others were heavily dependent on the source post/tweet to make sense.

## 3.2 Annotation Process

**Annotator Training** After the data was collected, it was annotated by two annotators, who underwent training to prepare themselves for the task. We considered 20 different propaganda techniques whose definitions can be found in Appendix A. Both annotators were experienced AI researchers with Master's degrees, fluent in English, Urdu, and Roman Urdu. Annotating propaganda techniques presents a greater challenge compared to tasks such as image classification or sentiment analysis. This is due to the annotators' requirement to comprehend each propaganda technique, recall and identify suitable techniques for specific texts, and accurately label the text spans associated with each propaganda technique. The training process consists of three stages. For Stages 1 and 2, both annotators were provided with English text examples containing propaganda. In Stage 1, both annotators were given a subset of these examples (Set A) and were asked to annotate them independently. The annotators had to identify and list all possible labels. Once they had completed their annotations, they compared their results and cross-questioned each other to resolve any conflicts that had arisen. After reconciling any discrepancies, they prepared a final document. Stage 2 is a repetition of Stage 1, but with a different set of examples. The annotators annotated Set B independently and prepared the document the same way they did for Set A. In Stage 3, the annotators discussed their annotated documents with a domain expert who provided valuable feedback to refine their understanding of propaganda techniques. After a discussion, both annotators updated their knowledge state and repeated the entire process, this time focusing on annotating the text spans and corresponding propaganda labels. This thorough training program ensured that the annotators had a comprehensive understanding of the propaganda techniques and were well-prepared to annotate the code-switched dataset.

Our approach is highly methodological, to ensure consistency across all examples in the dataset and to prevent any discrepancies that may impact the quality of our data. Figure 2 visually outlines the systematic process we followed while annotating the dataset. The annotation process begins with providing two annotators with an example from the dataset, which they must independently label by identifying the possible techniques used and their corresponding spans within the text.

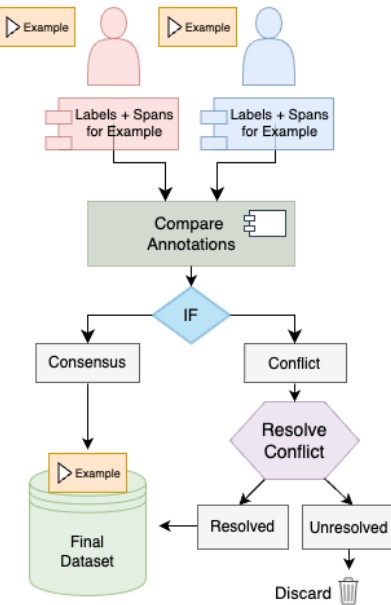

Figure 2: Our methodology for annotating the code-switched dataset with propaganda techniques.

Once the annotation was complete, the two annotators compared and consolidated their annotations. The consolidation process was done to ensure that there were no discrepancies in the labeling of techniques and spans. This step was crucial for ensuring that each example went through the same standard process. If both annotators agreed on the annotations, the example was approved and the labeled example was added to the dataset.

In the event of a conflict, where the two annotators had different labels or spans, they worked together to resolve that conflict. The conflict resolution process involved reviewing the propaganda class definitions that were under conflict, discussing the relevance of the propaganda class for that particular example, and cross-questioning each other to determine whether to include or to exclude the propaganda class in question. If consensus was reached, the example was approved and added to the dataset. However, if the conflict could not be resolved, the example was discarded. To ensure accuracy and to prevent fundamental errors, regular discussions between annotators and domain experts were conducted. These discussions were crucial to maintain the quality of the dataset, and they helped to identify potential errors during the annotation process. The main reason why two annotations of the same example would differ was primarily because one of the annotators missed a technique instance.

| Statistic | Value |
|---|---|
| # of Labeled Examples | 923 |
| # of Unlabeled Examples | 107 |
| Average Example Length | $147.56 \pm 53.79$ |
| Maximum Example Length | 400 |
| Minimum Example Length | 42 |
| Average Span Length | $63.64 \pm 67.81$ |
| Maximum Span Length | 400 |
| Minimum Span Length | 2 |
| Total # of Words | 28452 |
| Vocabulary Size | 7154 |

Table 1: Statistics about our dataset. Here the *unlabeled* examples are those with no propaganda class assigned.

There were rarely disagreements between the two annotators after an example with a conflict has been discussed. This process was repeated for all the examples in the dataset. Then, a quick double-checking over all examples was conducted. The purpose of this iteration was to ensure consistency in labeling the spans and the propaganda classes. In this iteration, the annotators were allowed to see the labels they made in the first iteration to ensure the uniformity of the decision-making process while labeling the spans and the propaganda classes.

To annotate code-switched text, we built a web-based platform. Snapshots of the interface are shown in Appendix B. Other work, like (Dimitrov et al., 2021a), used PyBossa[1] for annotation. Alternatives to PyBossa include Label Studio[2] and INCEpTION.[3] Although these annotation tools are powerful, we found some potential issues with them, such as limited customizability, insufficient documentation, a steep learning curve, or a need for a paid subscription.

### 3.3 Quality of the Annotations

To measure the quality of the annotations, we computed the Krippendorff's $\alpha$, a reliability co-efficient developed to measure agreement, which takes into account multi-label annotations (Artstein and Poesio, 2008). The Krippendorff's $\alpha$ score between the two annotators for our annotations is **0.76**. As per Krippendorff (2011), a score above 0.8 indicates a high level of agreement. Hence, our results demonstrate a fairly reasonable agreement between the annotators, particularly considering that the annotation involved multiple labels.

[1] https://pybossa.com/
[2] https://labelstud.io/
[3] https://inception-project.github.io/

| Propaganda Techniques | Number of Instances | Avg Span Length $\pm$ Std Dev. |
|---|---|---|
| Name calling/Labeling | 563 | $16.60 \pm 10.23$ |
| Repetition | 15 | $146.06 \pm 81.02$ |
| Doubt | 39 | $124.87 \pm 76.35$ |
| Reductio ad hitlerum | 8 | $119.00 \pm 19.51$ |
| Appeal to fear/prejudice | 87 | $144.13 \pm 50.20$ |
| Straw man | 18 | $141.38 \pm 53.94$ |
| Loaded language | 693 | $8.78 \pm 8.22$ |
| Bandwagon | 4 | $102.00 \pm 19.45$ |
| Smears | 382 | $144.25 \pm 60.03$ |
| Obfuscation, Int. vagueness... | 12 | $139.33 \pm 63.04$ |
| Glittering generalities (Virtue) | 44 | $91.97 \pm 55.74$ |
| Causal oversimplification | 86 | $90.96 \pm 42.44$ |
| Appeal to authority | 12 | $136.08 \pm 63.70$ |
| Red herring | 61 | $142.40 \pm 51.66$ |
| Thought-terminating cliché | 43 | $27.13 \pm 23.03$ |
| Black-and-white fallacy | 34 | $76.38 \pm 38.95$ |
| Slogans | 45 | $24.51 \pm 12.11$ |
| Whataboutism | 38 | $163.15 \pm 56.54$ |
| Exaggeration/Minimisation | 366 | $85.40 \pm 45.23$ |
| Flag-waving | 27 | $140.14 \pm 41.14$ |
| **Overall** | **2,577** | **$63.64 \pm 67.81$** |

Table 2: Total number of instances and average span lengths (number of characters) for each of the propaganda classes.

### 3.4 Statistics about the Dataset

Table 1 shows statistics about the dataset: it has a total of 1,030 examples with 2,577 labeled spans annotated with the use of 20 propaganda techniques at the span level.

In Table 2, we see statistics about the 20 classes with their total number of instances as well as the average span length $\pm$ the standard deviation. From the number of instances, it can be noted that there is a class imbalance, where the top-4 class instances comprise 77.6% of the total number of instances. These include *Loaded language*, *Exaggeration/Minimisation*, *Smears*, and *Name calling/Labeling*. The other 16 classes contribute less than 5% each. Table 2 shows the varying span lengths of different techniques.

Figure 3 shows a histogram of the number of techniques per example, i.e., the distribution of the number of distinct techniques with respect to the number of examples. For example, in the distribution, we can see $(x=2, y=263)$, which shows that there are 263 examples that contain **only** two distinct techniques. Another example $(x=0, y=107)$ shows that for 107 examples there are no techniques labeled. The maximum number of distinct techniques in an example is 9, which occurs only once in our entire dataset.

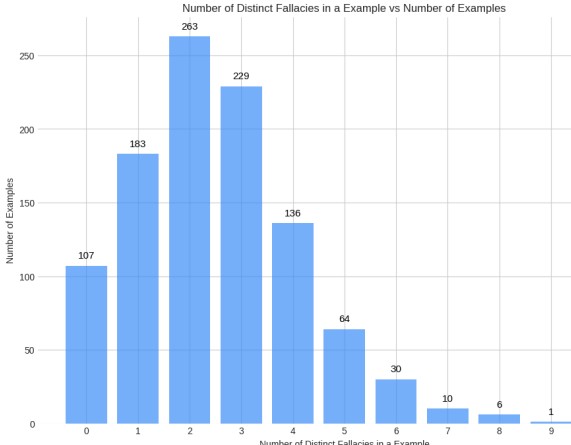

Figure 3: Histogram of the number of propaganda techniques per example.

| Fine-Tuning Strategy Type | Model | Model Name |
|---|---|---|
| Out of Domain Meme Dataset (Text-Only) | $\mathcal{M}_1$ | BERT |
| | $\mathcal{M}_2$ | mBERT |
| | $\mathcal{M}_3$ | XLM RoBERTa |
| Translated (Code-Switched → English) | $\mathcal{M}_4$ | BERT |
| | $\mathcal{M}_5$ | mBERT |
| | $\mathcal{M}_6$ | XLM RoBERTa |
| Code-Switched | $\mathcal{M}_7$ | BERT |
| | $\mathcal{M}_8$ | mBERT |
| | $\mathcal{M}_9$ | RUBERT |
| | $\mathcal{M}_{10}$ | XLM RoBERTa |
| | $\mathcal{M}_{11}$ | XLM RoBERTa (Roman Urdu) |
| | $\mathcal{M}_{12}$ | DeBERTaV3 |
| No fine-tuning | $\mathcal{M}_{13}$ | GPT-3.5-Turbo @20-shot |

Table 3: List of models and fine-tuning strategies (on particular datasets). Here $\mathcal{M}_*$ refers to a specific model fine-tuned using a specific fine-tuning *strategy type*.

## 4 Experiments

### 4.1 Models

In our experiments, we fine-tuned several state-of-the-art monolingual, multilingual, and cross-lingual pre-trained models on our labeled propaganda dataset. We use different fine-tuning strategies to adapt these models to our specific task. We also use Large Language Models (LLMs), specifically GPT-3.5 Turbo, by providing a few examples as prompts in different few-shot settings (see Appendix C for detail). Below are the pre-trained models we used.

**BERT** (Devlin et al., 2019) is a language model based on the Transformer architecture, trained on a large corpus of text using Masked Language Modeling (MLM) and Next Sentence Prediction (NSP).

**mBERT** (Devlin et al., 2019) is a multilingual pre-trained language model based on the BERT architecture that can represent words and phrases consistently across different languages by training on a multilingual corpus. It is trained on an extensive dataset of text in 104 languages.

**XLM RoBERTa** (Conneau et al., 2020) is a state-of-the-art pre-trained cross-lingual language model which is an extension of RoBERTa (Zhuang et al., 2021), pre-trained on large amounts of text data from over 100 different languages, using MLM and NSP pre-training objectives.

**RUBERT** (Khalid et al., 2021) is a bilingual Roman Urdu model created by continuing the pre-training of BERT on their own dataset as well as on other publicly available Roman Urdu datasets (Amjad et al., 2020; Azam et al., 2020; Khana et al., 2016).

**XLM RoBERTa (Roman Urdu)** (Aimlab, 2022) is fine-tuned on a Roman Urdu Twitter dataset (Khalid et al., 2021) without making any additions to the vocabulary, which is effective as the model attempts to learn a better contextual representation of the existing vocabulary.

**DeBERTaV3** (He et al., 2021) is an enhanced multilingual model, building upon the improvements made in DeBERTa (He et al., 2020), by replacing the MLM approach with Replaced Token Detection (RTD).

### 4.2 Fine-Tuning Strategies

We use three fine-tuning strategies for evaluating the performance of models on our code-switched dataset, as shown in Table 3. Models $\mathcal{M}_1$ to $\mathcal{M}_{12}$ are validated on our code-switched validation split, and their performances are evaluated on our code-switched test split. In the **Out-of-Domain** strategy, models $\mathcal{M}_1$ to $\mathcal{M}_3$ are fine-tuned on a meme propaganda dataset (Dimitrov et al., 2021a) using the text data extracted via OCR from the memes, without considering the image. For the **Translated (Code-Switched → English)** strategy, models $\mathcal{M}_4$ to $\mathcal{M}_6$ are fine-tuned on the English translated version of our code-switched training split. The translation is performed using GCP's Cloud Translation API,[4] which translates the text from Roman Urdu-English to English. Lastly, in the **Code-Switched** strategy, models $\mathcal{M}_7$ to $\mathcal{M}_{12}$ are fine-tuned on our code-switched training set. LLM $\mathcal{M}_{13}$ is used in a few-shot setting, where a limited number of code-switched training examples, along with their corresponding gold labels, are incorporated into the prompt.

---

[4]https://cloud.google.com/translate/docs/

| Fine-Tuning Strategy Type | Model | Avg. Precision | | Avg. Recall | | Avg. F1-Score | | Accuracy | Exact Match Ratio | Hamming Score |
|---|---|---|---|---|---|---|---|---|---|---|
| | | Micro | Macro | Micro | Macro | Micro | Macro | | | |
| Out of Domain Meme Dataset (Text-Only) | $\mathcal{M}_1$ | .57 | .16 | .18 | .05 | .27 | .07 | .898 | .083 | .185 |
| | $\mathcal{M}_2$ | .45 | .06 | .29 | .07 | .35 | .06 | .886 | .071 | .239 |
| | $\mathcal{M}_3$ | .44 | .07 | .33 | .08 | .39 | .07 | .889 | .083 | .261 |
| Translated (Code-Switched → English) | $\mathcal{M}_4$ | .45 | .12 | .44 | .12 | .44 | .10 | .884 | .038 | .288 |
| | $\mathcal{M}_5$ | .49 | .10 | .37 | .11 | .42 | .10 | .891 | .064 | .267 |
| | $\mathcal{M}_6$ | .54 | .26 | .40 | .14 | .46 | .16 | .900 | .103 | .320 |
| Code-Switched | $\mathcal{M}_7$ | .55 | .21 | .37 | .12 | .44 | .14 | .900 | .096 | .308 |
| | $\mathcal{M}_8$ | .50 | .24 | .32 | .12 | .39 | .14 | .893 | .083 | .263 |
| | $\mathcal{M}_9$ | .49 | .10 | .35 | .09 | .40 | .10 | .892 | .083 | .280 |
| | $\mathcal{M}_{10}$ | .54 | .21 | .43 | .16 | .48 | .17 | .901 | .110 | .354 |
| | $\mathcal{M}_{11}$ | **.59** | .34 | .49 | .22 | **.53** | .25 | **.910** | **.135** | **.375** |
| | $\mathcal{M}_{12}$ | .51 | **.53** | .43 | .15 | .46 | .17 | .895 | .090 | .307 |
| No fine-tuning | $\mathcal{M}_{13}$ | .39 | .31 | **.53** | **.42** | .45 | **.28** | .862 | .051 | .306 |

Table 4: Results on the nine evaluation measures listed in subsection 4.4 for the different models $\mathcal{M}_1$ to $\mathcal{M}_{13}$. Green highlights show the highest score for each of the evaluation measures.

## 4.3 Experimental Settings

We prepared our dataset by annotating propaganda labels at the fragment level, but we trained and evaluated our models in a multi-class multi-label setting without considering the span of the text. We used the PyTorch framework for our implementation and HuggingFace to load pre-trained state-of-the-art models. We took other pre-trained models from their source repositories. We ran each experiment on a single NVIDIA Quadro RTX 6000 24GB GPU. Due to GPU RAM limitation, we used a training and validation batch size of 12. Here are the remaining hyperparameter values: 10 epochs, maximum sequence of length 256, AdamW as the optimizer with epsilon 1e-8 and weight decay of 0.1, a learning rate of 3e-5 and Linear Warmup as the learning rate scheduler. We used the Binary Cross Entropy (BCE) as our loss function.

We used `iterative_train_test_split` from scikit-multilearn, which splits our multi-label data in a stratified manner, so that there is a balance of classes in our training, validation, and test splits. The training, validation, and test splits are 786 (76%), 89 (9%), and 155 (15%), respectively.

## 4.4 Evaluation Measures

We evaluated the performance of our models using micro/macro-average precision/recall/F1-score, hamming score, Exact Match Ratio (EMR), and accuracy. As we have a multi-label problem, i.e., each example can be assigned 0, 1, or multiple labels, with class imbalance, we use the micro-average F1-score as our primary evaluation measure.

## 5 Results

Table 4 shows the performance of different models on our code-switched test split. We can see that models $\mathcal{M}_{11}$ and $\mathcal{M}_{13}$ show relatively higher performance. We can further observe that $\mathcal{M}_{11}$, which was fine-tuned on Roman Urdu data using XLM-RoBERTa, demonstrates superior performance in terms of micro-average F1-Score, Accuracy, EMR, and Hamming Score. On the other hand, $\mathcal{M}_{13}$, GPT-3.5 Turbo, which is provided with 20 code-switched examples in the prompt, exhibits higher performance in terms of micro- and macro-average recall. Additional information regarding the performance of $\mathcal{M}_{13}$ on other few-shot settings can be found in Table C.1 in the Appendix C. Model $\mathcal{M}_{11}$ achieves a 5% higher micro-average F1-Score compared to the second-best model, $\mathcal{M}_{13}$. Moreover, $\mathcal{M}_{11}$ demonstrates a higher micro-average precision than its micro-average recall, implying that the model tends to produce fewer false positives. Essentially, the model is more conservative in its predictions and leans towards making fewer positive predictions overall. Additionally, $\mathcal{M}_{11}$ outperforms the other models in terms of Hamming score and EMR, which are two of the most commonly used evaluation measures for multi-label classification.

Table 5 shows the F1-Score for each model and class. The column *Percentage of Instances* indicates the percentage of instances for each propaganda technique in our test split, and we can observe that only four classes have a higher percentage than 5%.

| Models → 
 Propaganda Techniques ↓ | Percentage of Instances (%) | $\mathcal{M}_1$ | $\mathcal{M}_2$ | $\mathcal{M}_3$ | $\mathcal{M}_4$ | $\mathcal{M}_5$ | $\mathcal{M}_6$ | $\mathcal{M}_7$ | $\mathcal{M}_8$ | $\mathcal{M}_9$ | $\mathcal{M}_{10}$ | $\mathcal{M}_{11}$ | $\mathcal{M}_{12}$ | $\mathcal{M}_{13}$ |
|---|---|---|---|---|---|---|---|---|---|---|---|---|---|---|
| Loaded Language | 26.9 | .52 | .61 | .63 | .60 | .57 | .66 | .64 | .63 | .61 | **.74** | .70 | .70 | .63 |
| Obfuscation, Intentional vagueness, Confusion | 0.50 | .00 | .00 | .00 | .00 | .00 | .00 | .00 | .00 | .00 | .00 | .00 | .00 | .00 |
| Appeal to fear/prejudice | 3.40 | .00 | .00 | .00 | .12 | .00 | .30 | .20 | .30 | .00 | **.35** | .30 | .32 | .33 |
| Appeal to authority | 0.50 | .00 | .00 | .00 | .00 | .00 | .00 | .00 | .00 | .00 | .00 | .00 | .00 | .15 |
| Whataboutism | 1.50 | .00 | .00 | .00 | .00 | .14 | **.40** | .00 | .25 | .00 | .00 | .20 | .00 | .40 |
| Slogans | 1.70 | .00 | .00 | .00 | .00 | .00 | .00 | .00 | .00 | .00 | .00 | .00 | .00 | **.18** |
| Exaggeration/Minimisation | 14.2 | .00 | .00 | .00 | .25 | .17 | .29 | .40 | .31 | .44 | .47 | **.56** | .34 | .37 |
| Black-and-white Fallacy/Dictatorship | 1.30 | .00 | .00 | .00 | .00 | .00 | .00 | .29 | .25 | .10 | .00 | .33 | .33 | **.55** |
| Smears | 14.8 | .22 | .09 | .27 | **.59** | .57 | .57 | .47 | .47 | .48 | .49 | .53 | .48 | .40 |
| Doubt | 1.50 | .29 | .00 | .00 | .00 | .00 | .00 | .29 | .00 | .00 | .40 | **.50** | .22 | .31 |
| Bandwagon | 0.20 | .00 | .00 | .00 | .00 | .00 | .00 | .00 | .00 | .00 | .00 | .00 | .00 | **.50** |
| Name calling/Labeling | 21.8 | .32 | .46 | .52 | .51 | .57 | .52 | .52 | .32 | .45 | .51 | .63 | .56 | **.69** |
| Reductio ad hitlerum | 0.30 | .00 | .00 | .00 | .00 | .00 | .00 | .00 | .00 | .00 | .00 | .00 | .00 | **.12** |
| Presenting Irrelevant Data (Red Herring) | 2.40 | .00 | .00 | .00 | .00 | .00 | **.20** | .00 | .20 | .00 | .00 | .00 | .00 | .00 |
| Repetition | 0.60 | .00 | .00 | .00 | .00 | .00 | .00 | .00 | .00 | .00 | .00 | .00 | .00 | **.33** |
| Straw Man | 0.70 | .00 | .00 | .00 | .00 | .00 | .00 | .00 | .00 | .00 | .00 | .00 | .00 | .00 |
| Thought-terminating cliché | 1.70 | .00 | .00 | .00 | .00 | .00 | .00 | .00 | .00 | .00 | .00 | **.22** | .00 | .00 |
| Glittering generalities (Virtue) | 1.70 | .00 | .00 | .00 | .00 | .00 | **.29** | .00 | .00 | .00 | .25 | .20 | .22 | .20 |
| Flag-waving | 1.00 | .00 | .00 | .00 | .00 | .00 | .00 | .00 | .00 | .00 | .00 | **.36** | .00 | .22 |
| Causal Oversimplification | 3.30 | .00 | .00 | .00 | .00 | .00 | .00 | .00 | .13 | .00 | .22 | **.50** | .12 | .24 |

Table 5: Comparison of class-level performance (F1-Score) on 13 different models. The naming convention of the models can be found in Table 3. Green highlights indicate the highest F1-Score for each propaganda technique.

Comparing LLM $\mathcal{M}_{13}$ to the other fine-tuned models in Tables 4 and 5, we observe that although $\mathcal{M}_{13}$ does not achieve the highest micro-F1 score, it demonstrates significantly higher macro-average recall and macro-average F1 score. Moreover, it performs remarkably well on underrepresented classes, i.e., such with instances below 5%. This can be attributed to two main factors. First, LLMs have the advantage of leveraging their extensive pre-training and contextual understanding, which enables them to better comprehend and predict underrepresented classes, such as propaganda techniques in our case. This advantage sets them apart from smaller pre-trained models. Second, LLMs are trained on vast corpora containing diverse languages and code-switched text. This exposure enables them to capture general language patterns and knowledge, even on smaller low-resource or code-switched datasets. Consequently, the model exhibits a high macro-F1 score, indicating the LLM's ability to identify relevant instances for each class, regardless of the overall distribution of classes in the dataset.

Through various experiments, our study has revealed that training models on code-switched text yields superior performance for our propaganda detection task compared to training on translated versions of the code-switched text in high-resource languages. These findings emphasize the significance of fine-tuning the models specifically on code-switched text, as opposed to relying on the translated versions of the text in high-resource languages.

The potential reason for the lower performance that we observed in translating code-switched data to high-resource languages could be attributed to the translation service's inability to capture and preserve the inherent propagandistic characteristics present in code-switched text from low-resource languages. We also observe that the cross-lingual model's ability to transfer knowledge and capabilities from one language to another one allows it to exhibit superior performance compared to both monolingual and multilingual models when handling code-switched text.

## 6 Conclusion and Future Work

We proposed the novel task of detecting propaganda techniques in code-switched social media test. We created a corpus of 1,030 code-switched English and Roman Urdu texts annotated with the use of 20 propaganda techniques. We performed a number of experiments using different fine-tuning strategies and we found that modeling the multilinguality directly rather than using translation of the code-switched text yields better results.

In future work, we plan experiments with detecting propaganda techniques in code-switching text for other resource-poor languages. We further plan to expand our annotated corpus with many more examples. We also want to fine-tune pre-trained multilingual and cross-lingual language models specifically aiming at detecting fine-grained propaganda at the fragment level. Last but not least, we want to understand the best strategy for handling code-switched text with LLMs.

## Limitations

### Language

Our first limitation is related to the language differences between Roman Urdu and English, which we faced while annotating code-switched propaganda text. This limitation stemmed from the fact that the original annotations for fine-grained propaganda techniques started with English, and thus the annotation guidelines reflected the English language and grammar. However, due to the different linguistic typologies of Urdu and English, we found it difficult to choose text spans that mix Roman Urdu and English. English follows a subject–verb–object (SVO) sentence structure, where the subject comes first, the verb is second, and the object is third. Urdu, on the other hand, follows a subject–object–verb (SOV) sentence structure where the subject comes first, the object is second, and the verb is third. Another constraint posed by the languages is that Roman Urdu is written in Latin/Roman script. Therefore, when individuals write Urdu in Roman Urdu, there is a variability in the spellings for some words. This causes problems as two different spelling will identify as two unique words in our model dictionary, where in fact they might actually correspond to the same word. Moreover, we use code-switched translations to English, but translating propaganda text between languages presents a challenge because propaganda techniques may be language-specific and thus difficult to preserve in translation. This results in texts losing their propagandistic nature in translation, which makes it difficult to compare propaganda detection across languages.

### Dataset

We faced several constraints related to the small size of the dataset and the class imbalance of the propaganda labels. These limitations result in various types of noise in the dataset, including random noise and systematic bias. In small datasets, the model may seem to perform well on the training and on the testing splits, but this may not accurately reflect its performance on new data. This is because the small dataset may not represent the broader range of data that the model may encounter, resulting in unreliable evaluation results. Class imbalance can also have a significant impact on evaluation and may result in biased performance results, which makes the evaluation measures noisy.

### Annotation

Annotating propaganda techniques at the fragment level differs from most other annotation tasks, as it is an very time-consuming and mentally demanding. We assigned two annotators who underwent comprehensive training to accurately annotate propaganda within the text. By engaging in extensive training sessions and analyzing pre-labeled examples from domain professionals, the annotators became proficient in identifying propaganda in mixed Roman Urdu and English code-switched text. However, it is essential to acknowledge the limitation regarding the number of volunteers willing to undergo such training. Additionally, among those who are willing to participate in the training, there is a scarcity of individuals who possess fluency in both Urdu and English languages, along with a solid understanding of Roman Urdu (i.e., Urdu written using the Latin script).

### Definition and Number of Propaganda Techniques

Another limitation is differing views on the exact number of propaganda techniques. While experts generally agree on the definition of propaganda, there are noticeable discrepancies in the techniques listed by various scholars (Torok, 2015). These differences occur due to one of two reasons. A scholar may ignore a set of techniques while compiling their list, or may use broader definitions that cover more fine-grained definitions used by other scholars. For instance, Miller (1939) considers a total of seven techniques, whereas Weston (2018) proposes 24 propaganda techniques, while Wikipedia[5] lists over 70 techniques.

## Ethics and Broader Impact

### User Privacy

The dataset we have only consisted of code-switched social media text and the social media platform it was taken from. It does not contain any user information. The data collectors refrained from recording any user names or links, to maintain complete anonymization. The data collection was limited to public accounts, pages, and groups, and by adhering to this restriction, the collected data aligned with the platforms' public data policies, allowing it to be used for public purposes.

---

[5]https://en.wikipedia.org/wiki/Propaganda_techniques

**Biases**

We acknowledge that the process of annotating propaganda techniques can involve subjectivity, which inevitably leads to biases in our gold labels or the distribution of the labels. To address these concerns, we first collect examples from multiple diverse users. We also ensure that rigorous annotation training is conducted for the annotators. Finally, we use a well-defined annotation process with a clear methodology. The high agreement score between our annotators gives us confidence that the labels assigned to the data are accurate in the majority of cases.

**Misuse Potential**

We kindly request researchers to be aware of the potential for malicious misuse of our dataset. Consequently, we urge them to exercise caution.

**Data Collector and Annotator Rights**

The involvement in the data collection and annotation was entirely voluntary. An informed consent was obtained from all participants before data collection, and their identities were protected through anonymization. They had the option to withdraw their consent at any time without any penalty. Researchers were transparent about the purpose of the annotation and the intended use of the annotated data with the participants. The risks to the participants were very low and comparable to what one encounters in their daily life. Last, all participants received fair monetary compensation for their participation, which was well above the minimal wage in their country.

**Intended Use**

The aim of our dataset is to encourage the detection of social media content with propagandistic characteristics, particularly in code-switched text. We firmly believe that this resource holds significant value for the research community, particularly when used appropriately and effectively.

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

# Appendix

## A   Propaganda Techniques

In this section, we go over the formal definitions for each of the propaganda techniques. These definitions are sourced from Dimitrov et al. (Dimitrov et al., 2021a), with some modifications made for clarity. The previous work used 22 propaganda techniques, but we only use 20 for our dataset annotation. We decided to exclude *Transfer* and *Appeal to Emotions* as they were only applicable to the images in the memes and were not relevant to our code-switched text. Additionally, we provide code-switched examples for each technique. The underlined text in the English translation shows the fragments that is associated with the propaganda technique.

### 1. Loaded Language

Loaded language refers to the use of words or phrases with strong emotional indications (positive or negative). It is used to influence the public's opinion or mindset by using these words or phrases with strong connotations.
**English example:** *"how stupid and petty things have become in Washington"*
**Code-switched example:** *"I hope women get over their bad boy syndrome soon. Pehle ameer baap ki aulaad dekh ke shaadi krti hain phir domestic abuse ke randi rone, you girls know what you're getting yourself into"*
**English translation:** *"I hope women get over their bad boy syndrome soon. First they see rich dads kids and marry them and then cry foul of domestic abuse, you girls know what you're getting yourself into"*

### 2. Name Calling/Labeling

Name Calling/Labeling refers to labeling the object of propaganda campaign (individual or group) as something that the public lovers, praises or admires or something that they find undesirable, fear and hate. Primarily they are in the form of an argument where labels are insulting or demeaning.
**English example:** *"Republican congressweasels"*, *"Bush the Lesser"*
**Code-switched example:** *"Patwaari aur youthiye larte rehein ge, meanwhile Nawaz and Niyazi will loot the country and leave for England"*
**English translation:** *"Patwaari and youthiye will keep on fighting, meanwhile Nawaz and Niyazi will loot the country and leave for England"*

### 3. Repetition

Repetition refers to repeating the same message multiple times so that it is eventually accepted among the public. Repetition makes a fact seem more true, regardless of whether it is or not.
**English example:** *"I promise, I promise, I promise I will deliver."*
**Code-switched example:** *"If they were previously politically motivated then in saroon ka court marshall kuro. Aur saza dou saza dou saza dou saza dou"*
**English translation:** *"If they were previously politically motivated then have them all court martialled. And punish them punish them punish them punish them"*

### 4. Exaggeration/Minimization

Exaggeration refers to overstating an idea, product's value or someone's characteristics than what they actually are to propagate their own narrative. This technique involves "hyping" up by using impressive sounding words that are nonetheless meaningless and vague. Minimization on the other hand refers to understating an idea, product's value or someone's characteristics than what they actually are. This is particularly done to make something seem less important or smaller than it is.
**English example:** *"I am so hungry I could eat a horse."*
**Another English example:** *"I did not hit her; it was a slight pat"*
**Code-switched example:** *"You are the top crypto scammer of Pakistan. Tumhain pata hai tum ne in masoom logon ke sath kya kya kiya hai kabhi ponzi scheme, kabhi courses aur kabhi paid subscriptions"*
**English translation:** *"You are the top crypto scammer in Pakistan. You know what you have done to these innocent people - sometimes running Ponzi schemes, other times offering courses or paid subscriptions."*

### 5. Doubt

Doubt refers to questioning the credibility of something or someone. Some people use it to cast doubt on their opponents, controversial issues or the credibility of some media organizations.
**English example:** *"A candidate talks about his opponent and says: Is he ready to be the Mayor?"*
**Code-switched example:** *"Death penalty is ridiculous. Aik rapist ko latka dene se baqion ko kia faraq*

*pare ga?"*
**English translation:** *"Death penalty is ridiculous. What difference will it make to others if one rapist is hanged?"*

## 6. Appeal to Fear/Prejudice

Appeal to Fear/Prejudice refers to seeking to create support for an idea by instilling anxiety and/or attempting to increase fear in the population toward an alternative. In some cases, the support is built based on preconceived judgements and acts as a tactic commonly used in marketing, politics, and media.
**English example:** *"We must stop these refugees; they are terrorists"*
**Code-switched example:** *"Pakistan ne bankrupt hojana soon. Pack your bags aur nikal lo Canada before it's too late"*
**English translation:** *"Pakistan is about to get bankrupt soon. Pack your bags and leave for Canada before it's too late"*

## 7. Appeal to Authority

Appeal to Authority refers to stating that a claim must be true simply because a valid authority or expert on the issue said it was true, without any other supporting evidence(Goodwin and McKerrow, 2011).
**English example:** *"Richard Dawkins, an evolutionary biologist and perhaps the foremost expert in the field says evolution is true, therefore it must be true."*
**Code-switched example:** *"Imran Khan keh raha hai isko follow nai karna chahye tou isko unfollow karna is the best option"*
**English translation:** *"Imran Khan is saying that we shouldn't follow him so unfollowing him is the best option"*

## 8. Flag-Waving

Flag-Waving refers to the technique which plays on strong national feeling to justify an action based on the undue connection to nationalism or patriotism or benefit for an idea, group or country.
**English example:** *"Entering this war will make us have a better future for our country."*
**Code-switched example:** *"Afghanion ko Pakistan se nikalo, haraamkhor humara khaa kr humein hi gaali dete hain. Also remember how they caused destruction through Kalashnikov culture in our beloved country"*
**English translation:** *"Expel Afghanis from Pakistan, these morally corrupt people benefit from our land and then curse us. Also remember how they caused destruction through Kalashnikov culture in our beloved country"*

## 9. Causal Oversimplification

Causal Oversimplification assumes a single cause or reason when there are actually multiple causes for an issue. This includes transferring blame to one person or group of people without investigating the complexities of the issue.
**English example:** *"The reason New Orleans was hit so hard with the hurricane was because of all the immoral people who live there."*
**Code-switched example:** *"Women divorce whenever they need money. Muft mein alimony bhi lo bachon ki custody bhi"*
**English translation:** *"Women divorce whenever they need money. Get the alimony for free and the children's custody"*

## 10. Slogans

Slogans refer to a brief, striking phrase that may include labeling and stereotyping which acts as an emotional appeal. This technique attempts to arouse prejudices in an audience by labeling the object as something the target audience likes or dislikes (Dan, 2015).
**English example:** *"Make America great again!"*
**Code-switched example:** *"Utho Pakistanio! Save your country!"*
**English translation:** *"Wake up Pakistanis! Save your country!"*

## 11. Black-and-White Fallacy/Dictatorship

Black-and-White Fallacy presents two opposite/alternative options as the only possibilities, when in fact more possibilities exist (Torok, 2015). Dictatorship is an extreme case where the audience is told exactly what actions to take, eliminating any other possible choices.
**English example:** *"Either we go to war or we will perish"*
**Code-switched example:** *"Either you are a Muslim woman or a feminist, koi beech ka raasta nhi, faisla kr lo and stick to it"*
**English translation:** *"Either you are a Muslim woman or a feminist, there is no middle way, decide now and stick to it"*

## 12. Thought-terminating cliche

Words or phrases that discourage critical thought and meaningful discussion about a given topic. They are typically short, generic sentences that offer seemingly simple answers to complex questions or that distract the attention away from other lines of thought (Hunter, 2015).

**English example:** *"It is what it is", "It's common sense", "It doesn't matter", "You can't change human nature"*

**Code-switched example:** *"No matter how much you try, mard aur orat kabhi barabar nhi ho sakte. It is what it is"*

**English translation:** *"No matter how much you try, man and woman can never be equal. It is what it is"*

## 13. Whataboutsim

Whataboutism refers to when a critical question or argument is not answered or discussed, but deflected with a critical counter-question that expresses a counter-accusation. Here an opponent's position is discredited by charging them with hypocrisy without directly disproving their argument (Richter, 2017).

**English example:** *"A nation deflects criticism of its recent human rights violations by pointing to the history of slavery in the United States"*

**Code-switched example:** *"Jahez ke naam pe itni takleef kyu hoti hai aurton ko, what about their demands of big house and big car from the husband?"*

**English translation:** *"Why do women feel agitated in the name of dowry, what about their demands of big house and big car from the husband?"*

## 14. Reductio ad Hitlerum

Reductio ad Hitlerum can be thought of as playing the Nazi card. It is an attempt to persuade the audience to disprove an action or idea of someone by invalidating their position on the basis that their view is popular among hated groups. It can refer to any person or concept with a negative connotation (Teninbaum, 2009).

**English example:** *"Such a thought would come in the mind of Hitler"*

**Code-switched example:** *"This movie has been directed by a gay guy iss movie ko phailainay ke bajai isko rokna chahye"*

**English translation:** *"This movie has been directed by a gay guy rather than spreading this movie they should stop it's spread."*

## 15. Presenting Irrelevant Data (Red Herring)

Red Herring refers to introducing irrelevant material to the issue being discussed so that everyone's attention is diverted away from the points made. This form of misleading or distraction may either be a logical fallacy or literary device. Those subjected to a red herring argument are led away from the issue that had been the focus of the discussion and urged to follow an observation or claim that may be associated with the original claim, but is not highly relevant to the issue in dispute (Teninbaum, 2009).

**English example:** *"You may claim that the death penalty is an ineffective deterrent against crime – but what about the victims of crime? How do you think surviving family members feel when they see the man who murdered their son kept in prison at their expense? Is it right that they should pay for their son's murderer to be fed and housed?"*

**Code-switched example:** *"Joyland is banned in Punjab but is playing in cinemas across Sindh. Achi tarah pta chal gaya kon si hakumat ghulamon ki hai"*

**English translation:** *"Joyland is banned in Punjab but is playing in cinemas across Sindh. We now know very well whose government is being run by slaves"*

## 16. Bandwagon

Bandwagon refers to the fallacious argument which is based on claiming a truth or affirming something is good because the majority thinks so. It is used to persuade the target audience to join in and take the course of action because "everyone else is taking the same action"(Hobbs and McGee, 2014).

**English example:** *"Would you vote for Clinton as president? 57% say yes"*

**Code-switched example:** *"We hate Babar cause all of India does. Apnay liay nai tou issiliay karlo ke hum sab bhi kartay hain"*

**English translation:** *"We hate Babar cause all of India does. If not for yourself you should hate him because we all do."*

## 17. Obfuscation, Intentional vagueness, Confusion

Obfuscation, Intentional vagueness, and Confusion can be termed as deliberately making use of unclear words or phrases so that the target public create their own interpretation (Weston, 2018). For

instance, when an unclear phrase with multiple possible meanings is used within the argument, and, therefore, it does not really support the conclusion.

**English example:** *"It is a good idea to listen to victims of theft. Therefore, if the victims say to have the thief shot, then you should do it."*

Here the word "listen" can either mean to listen to the personal account of the experience of being a victim of theft and empathize with them or it could mean to carry out the punishment of their choice.

**Code-switched example:** *"As we all know in big guns circles a term used "Get a promotion with wife swap". Baqi aap samajhdar ho keh keya matlab hey es ka"*

**English translation:** *"As we all know in big guns circles a term used "Get a promotion with wife swap". The rest of you are wise enough to understand the meaning of this"*

### 18. Misrepresentation of Someone's Position (Strawman)

Strawman refers to refuting of an argument where the real subject of the argument was not addressed or refuted but instead substituted with a weaker similar one in place of the original so that it can be easily confuted. One who engages in this fallacy is said to be "attacking a straw man".

**English example:** *"Claiming that all vegans are opposed to all forms of animal captivity, including pet ownership."*

**Code-switched example:** *"Aurat march is trash as rather than helping women who actually needs helps all they talk about is nanga honei do, lesbian honei dou"*

**English translation:** *"Aurat march is trash as rather than helping women who actually needs helps all they talk about is let us naked, let us be lesbians"*

### 19. Glittering generalities (Virtue)

Glittering generalities are vague "feel good" statements that people are predisposed to want to identify with. The technique makes use of emotionally appealing phrases or words that are closely associated with highly valued concepts and beliefs such that they carry conviction without supporting information or reason. Such highly valued concepts attract general approval when attached to a person or issue. For example: Peace, hope, happiness, security, leadership, freedom, family values, and patriotism.

**English example:** *"Because I'm Worth It (L'Oreal makeup)", "I'm Lovin' It (McDonald's fast food)", "The Best a Man Can Get (Gilette razors)"*

**Code-switched example:** *"Taliban are the true mujahids defending their country like lions"* **English translation:** *"Taliban are the true warriors defending their country like lions"*

### 20. Smears

A smear refers to an effort to damage or call into question someone's reputation, by propounding negative propaganda through false accusations and slander, etc. It can be applied to individuals or groups.

**English example:** *"I'm not pregnant, I haven't been tested. But the President says if it's not tested it's not real. I think it's gas..."*

**Code-switched example:** *"Malik Riaz is a cult. PPP ke saath mil kar paani ke daamon zamin kharredi hai to make Bahria Town Karachi"*

**English translation:** *"Malik Riaz is a cult. After colluding with PPP, land property was bought at extremely low prices to make Bahria Town Karachi"*

## B  Annotation Web Platform

In this section, we present our web-based annotation platform, which we developed from scratch to annotate our code-switched dataset. The basic layout of our website is presented in Figure B.1. The right side of the page displays a list of propaganda techniques, each with a corresponding colored box. On the left side, there is a text box where code-switched text is annotated. The text is imported from a comma separated values (CSV) file with columns names: ID, TEXT, LABELS, INCLUDED, and TRANSLATED. In Figure B.2, we can see the annotation feature which is clearly shown as a 2-step process. Here, we can see that the text example is already labeled as *Black-and-White Fallacy/Dictatorship* in the JSON output. To label another span of text, we follow two steps. In Step 1, we select the span of text that needs to be labeled. In Step 2, we click on the colored boxes corresponding to the label we wish to assign to that span. In Figure B.2, the first step illustrates the selection of the span "Aurat Raj na manzoor", while the second step displays the updated annotation in JSON format below, where the span is labeled as a *Slogan*.

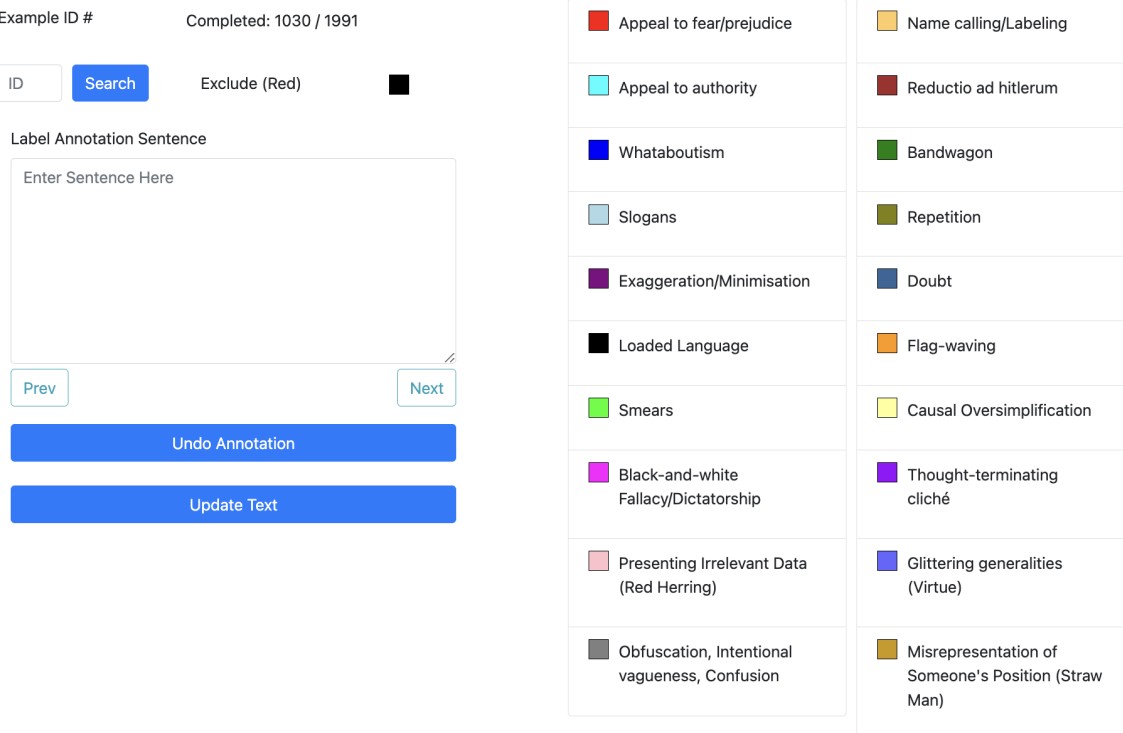

Figure B.1: Interface design of our web-based annotation tool.

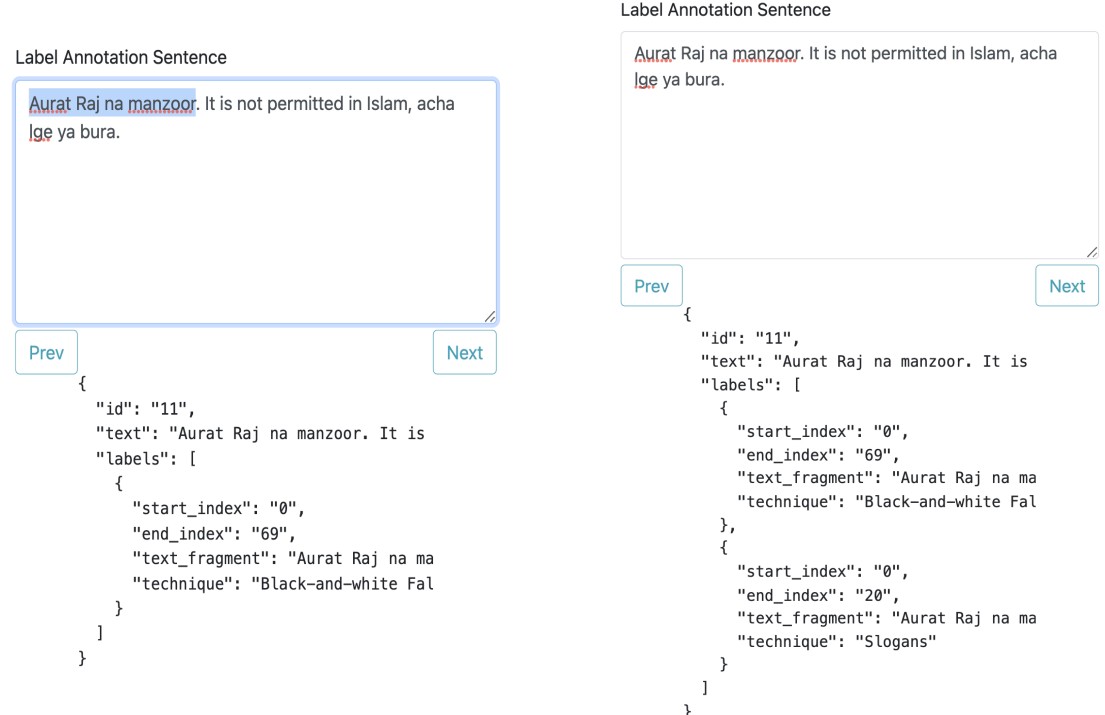

Figure B.2: The two-step annotation process of a text. Step 1: Select the span of text that needs to be annotated. Step 2: Click on the boxes next to the corresponding propaganda label shown in Figure B.1. The image on the right shows the updated JSON output after completing the two steps.

## C  Prompts for Propaganda Detection using GPT-3.5-Turbo

In this section, we leverage the power of Large Language Models (LLMs), specifically `GPT-3.5 Turbo`, to enhance our language processing capabilities. In particular, we use LLMs by providing them with a set of examples within the prompt and run them in four different few-shot settings. The prompt incorporates a small set of code-switched training examples, along with their respective gold labels, to facilitate the LLM in generating accurate predictions for the code-switched test set. Table C.1 shows their results on several different evaluation measures.

An example of a 10-shot prompt is shown below:

```
"""
    Can you help me analyze a code-switched text in Roman Urdu and English to
    identify one or multiple propaganda techniques? There can also be no propaganda
    techniques in a text.
    The list of 20 propaganda techniques is given below:

    - Loaded Language
    - Obfuscation, Intentional vagueness, Confusion
    - Appeal to fear/prejudice
    - Appeal to authority
    - Whataboutism
    - Slogans
    - Exaggeration/Minimisation
    - Black-and-white Fallacy/Dictatorship
    - Smears
    - Doubt
    - Bandwagon
    - Name calling/Labeling
    - Reductio ad hitlerum
    - Presenting Irrelevant Data (Red Herring)
    - Repetition
    - Misrepresentation of Someone's Position (Straw Man)
    - Thought-terminating cliche
    - Glittering generalities (Virtue)
    - Flag-waving
    - Causal Oversimplification

    Here are a few examples of different code-switched texts, along with their
    output in the form of comma-separated propaganda techniques, provided for your
    guidance.

    Examples:

        Text: This movie has been directed by a gay guy iss movie ko phailainay ke
    bajai isko rokna chahye. They are just manipulating the whole situation. Also it
     won the queer award.
        Output: Reductio ad hitlerum, Smears, Loaded Language, Exaggeration/
    Minimisation, Appeal to fear/prejudice

        Text: Joyland is banned in Punjab but is playing in cinemas across Sindh.
    Achi tarah pta chal gaya kon si hakumat ghulamon ki hai.
        Output: Presenting Irrelevant Data (Red Herring), Loaded Language

        Text: We hate Babar cause all of India does. Apnay liay nai tou issiliay
    karlo ke hum sab bhi kartay hain
        Output: Bandwagon, Exaggeration/Minimisation, Loaded Language

        Text: Aurat march is trash as rather than helping women who actually needs
    helps all they talk about is nanga honei do, lesbian honei dou
        Output: Misrepresentation of Someone's Position (Straw Man), Smears, Loaded
    Language, Name calling/Labeling, Exaggeration/Minimisation
```

Listing 1: Example of 10-shot ChatGPT (GPT-3.5 Turbo) Prompt (1/2).

```
"""
      Text: Either you are a Muslim woman or a feminist, koi beech ka raasta nhi,
   faisla kr lo and stick to it
      Output: Black-and-white Fallacy/Dictatorship

      Text: No matter how much you try, mard aur orat kabhi barabar nhi ho sakte.
   It is what it is
      Output: Thought-terminating cliche

      Text: I hope women get over their bad boy syndrome soon. Pehle ameer baap ki
   aulaad dekh ke shaadi krti hain phir domestic abuse ke randi rone, you girls
   know what you're getting yourself into.
      Output: Appeal to fear/prejudice, Name calling/Labeling, Loaded Language,
   Causal Oversimplification

      Text: Patwaari aur youthiye larte rehein ge, meanwhile Nawaz and Niyazi will
   loot the country and leave for England.
      Output: Name calling/Labeling, Loaded Language, Smears

      Text: Women divorce whenever they need money. Muft mein alimony bhi lo
   bachon ki custody bhi
      Output: Loaded Language, Causal Oversimplification, Exaggeration/
   Minimisation

      Text: Taliban are the true mujahids defending their country like lions
      Output: Glittering generalities (Virtue), Name calling/Labeling

      The Text to analyze is: {}
      Please make sure the answer is comma seperated if multiple propaganda
   techniques are given.
"""
```

Listing 2: Example of 10-shot ChatGPT (GPT-3.5 Turbo) Prompt (2/2).

| Model GPT-3.5 Turbo | Avg. Precision | | Avg. Recall | | Avg. F1-Score | | Accuracy | Exact Match Ratio | Hamming Score |
|---|---|---|---|---|---|---|---|---|---|
| | Micro | Macro | Micro | Macro | Micro | Macro | | | |
| 0-shot | **.43** | **.39** | .25 | .12 | .31 | .12 | **.884** | **.083** | .214 |
| 5-shot | .35 | .33 | .53 | .35 | .42 | .22 | .846 | .032 | .279 |
| 10-shot | .37 | .30 | **.54** | .41 | .44 | .25 | .853 | .032 | .299 |
| 20-shot | .39 | .31 | .53 | **.42** | **.45** | **.28** | .862 | .051 | **.306** |

Table C.1: Performance evaluation metrics for different shot settings using the GPT-3.5 Turbo model. The highest scores are in bold and highlighted in green .