# OpenReview forum: "Detecting Propaganda Techniques in Code-Switched Social Media Text"
_EMNLP/2023/Conference — EMNLP 2023 Main_

### Official Review · Reviewer_YUyd · 2023-08-03

**Soundness:** 2

**Excitement:**

2: Mediocre: This paper makes marginal contributions (vs non-contemporaneous work), so I would rather not see it in the conference.

**Missing References:**

Rather than missing, some references include wrong information. For instance:

Barrón-Cedeno -> Barrón-Cedeño
G Martino -> Giovanni Da San Martino
Goodwin and Raymie McKerrow seems incomplete.

**Paper Topic And Main Contributions:**

The manuscript introduces a new corpus for propaganda identification at the span level for code-switched text Urdu-English. Then experiments are performed, at a different granularity level.

**Questions For The Authors:**

Question A: Authors claim that "one main method of circulating falsified information is through propaganda". This is not true. Propaganda does not necessarily involve false information.

Question B: How did the authors get permission to publish data coming from Facebook and Instagram?

Question C: Section 3.1. What are the criteria to select a post? What type of post/from what sort of account is considered?

Question D: line 292. What is a domain expert here? Why asking AI experts to annotate instead of domain experts?

Question E: "if the conflict cannot be resolved, the example is discarded." This is unclear. Is a whole post discarded in the presence of one single disagreement?

Question F: the authors developed an annotation interface because they claim that existing alternatives run short. Are they going to release it?

Question G: Table 1. The length is measured in terms of tokens or characters? If the largest span contains 400 tokens (or even characters), it sounds huge. What is such a long span?

Question H: Table 3. How is this translated? What is the quality of such translation?

Question I: "i.e., each example can be assigned zero, one, or multiple labels" What is an example? The whole post?

**Reasons To Accept:**

This seems to be the first effort to produce resources for propaganda techniques identification in code-switched Urdu-English text.

**Reasons To Reject:**

The authors claim this is a new task, and even mention it as one of the main contributions, but it is not. It is still propaganda technique identification, it is just that the text is not monolingual.

The authors annotate spans and multi-label, but their experiments neglect the spans. This is counterintuitive and unrealistic.

**Reproducibility:**

4: Could mostly reproduce the results, but there may be some variation because of sample variance or minor variations in their interpretation of the protocol or method.

**Reviewer Confidence:**

4: Quite sure. I tried to check the important points carefully. It's unlikely, though conceivable, that I missed something that should affect my ratings.

**Typos Grammar Style And Presentation Improvements:**

"During the American Revolution in the 18th century, newspapers and the printing press in various American colonies propagated their views to promote patriotism"

There were many American colonies, from Canada to Argentina. Most of them turned independent in the 19th century. I guess what the authors mean by American Revolution is the War of Independence of the 13 colonies against the UK.

"Rashkin et al. (2017) 170 focused on detecting multi-class propaganda at a 171 document-level on their annotated corpus". This is not true. It is multi-class, but not multi-class propaganda. Propaganda is just one of the four classes.

Table 2 does not demonstrate. It shows. The same applies to M11 and M13, Roberta. They do not demonstrate. They show.

The fontsize of the number sin Fig. 3 should be larger.

---

> ### Author Rebuttal · Authors · 2023-08-29
>
> **Reasons to Reject**
>
> ---
> ---
>
> **Comment** The authors claim this is a new task, and even mention it as one of the main contributions, but it is not. It is still propaganda technique identification, it is just that the text is not monolingual.
>
> **Response** We appreciate the reviewer's feedback and would like to clarify the distinction between our proposed task and existing propaganda technique identification. While it's true that both tasks involve identifying propaganda techniques, our task of detecting propaganda techniques in code-switched text introduces novel challenges. In existing propaganda detection, the focus is primarily on the monolingual text which is in a high-resource language i.e., English. In contrast, our task deals with code-switched text, which combines two languages (English and Roman-Urdu) including a low-resource language where no work has been done. This is a different domain and requires a different dataset and approach to tackle problems such as complexities related to language switching and cultural nuances that are typically not present in monolingual text.
>
> ---
>
> **Comment** The authors annotate spans and multi-label, but their experiments neglect the spans. This is counterintuitive and unrealistic
>
> **Response** We recognize that our data is annotated at the fragment/span level, while the models were trained on example-level classes, specifically propaganda techniques. Our primary objective in this work was to introduce a code-switched (English + Roman Urdu) dataset annotated with propaganda techniques at the fragment level. Nevertheless, we have presented preliminary results of the detection of propaganda techniques in the entire example, without predicting the spans of the techniques. Our plan is to enhance this research by extensively analyzing span/fragment-level propaganda techniques using pre-trained language models.
>
> A key factor affecting our decision to avoid fragment-level detection initially was our preliminary analysis, which mainly involved comparing our outcomes with those from out-of-domain and code-switched texts translated to English. Translating our code-switched data into English would inevitably alter the propaganda technique spans due to differing sentence structures—Urdu (Roman Urdu) follows a subject-object-verb (SOV) structure, while English follows subject-verb-object (SVO). Hence, this structural distinction would require us to do manual annotation of the translated text. Furthermore, the out-of-domain text is also annotated as a multi-label multi-class challenge and would also require manual annotation for that dataset.
>
>
> &nbsp;
> &nbsp;
> &nbsp;
> &nbsp;
> &nbsp;
>
> **Questions for the Authors**
>
> ---
> ---
>
>
> **Question-A** Authors claim that "one main method of circulating falsified information is through propaganda". This is not true. Propaganda does not necessarily involve false information.
>
> **Response** We agree with the reviewer that propaganda is not exclusively tied to false information. However, our assertion doesn't claim that all propaganda hinges on false data; rather, we emphasize it as one method among various others. Our definition, as stated in lines 41 to 43, clarifies: "Propaganda is the dissemination of biased or misleading information in order to manipulate people's beliefs and opinions towards a particular objective." In this context, "biased" signifies an inclination toward a specific perspective, resulting in an unfair representation of information. This information is frequently true but skewed. "Misleading" refers to information, statements, or actions that create a false impression or deceive, even when the underlying statement is accurate.
> Moreover, among the propaganda techniques we examine is "Loaded Language," which employs emotive language to evoke strong feelings. This technique doesn't necessarily involve false information. Many other techniques, like this one, also don't disseminate false information.
>
> ---
>
> **Question-B** How did the authors get permission to publish data coming from Facebook and Instagram?
>
> **Response** We collected public data by following Facebook and Instagram’s policy on “Public Information”. While collecting data, we ensured that data privacy related concerns are addressed. We have not included source links or user IDs in the data for each of the code-switched examples/texts. We collected the data from posts that fall under the category of “Public Information” according to Facebook and Instagram data policies. All users of these social media sites agree to the terms and conditions regarding the information they post on them publicly. We did not collect data from any groups which were not in public view.
>
> ---
>
> **Question-C** Section 3.1. What are the criteria to select a post? What type of post/from what sort of account is considered?
>
> **Response** We chose those posts/texts that we found were richer in the propaganda techniques so that we could capture more propaganda techniques in our limited data. We also ensure we select posts/texts based on the percentage of Roman Urdu and English words making sure that there are enough words from each language. Having said that, we also added posts/texts/examples which had no labels present. The social media texts are collected from the public Facebook accounts of the 4 data collectors, and we also followed various public Facebook groups/pages on different topics, such as politics, cricket, feminism, women’s march (Aurat march), religious views, gender equality, and more to collect text from diverse topics.
>
>
> ---
>
> **Question-D** Line 292. What is a domain expert here? Why asking AI experts to annotate instead of domain experts?
>
> **Response** When referring to a "Domain Expert," we mean an individual who possesses prior experience supervising the labeling of text into various propaganda techniques, including those employed in our study. We want to clarify that the term "AI experts" denotes that both annotators were actively engaged in AI research and were not intended to favor them over domain experts. Both annotators were proficient in English, Urdu, and Roman-Urdu languages. They underwent training under the guidance of the main "Domain Expert" to annotate the data. Following training, both annotators became qualified and well-prepared to annotate code-switched data. Finding existing domain experts in this field proved challenging, as few individuals were well-acquainted with identifying the labels and spans of the propaganda techniques we employed. This challenge was further compounded when narrowing down the pool to those who are familiar with native Urdu, Roman Urdu, and English languages.
>
>
> ---
>
> **Question-E** "if the conflict cannot be resolved, the example is discarded." This is unclear. Is a whole post discarded in the presence of one single disagreement?
>
> **Response** We apologize for any confusion caused by our statement. To clarify, if a conflict arises regarding any of the fragments or spans within an example, and both authors are in disagreement over the identified propaganda, the entire example or post is discarded.
>
> ---
>
> **Question-F** The authors developed an annotation interface because they claim that existing alternatives run short. Are they going to release it?
>
> **Response** We plan to make the code of our newly developed annotation platform, which annotates propaganda at the fragment level, publicly available. Additionally, we intend to deploy our annotation platform as a demo project on a web service.
>
> ---
>
> **Question-G** Table 1. The length is measured in terms of tokens or characters? If the largest span contains 400 tokens (or even characters), it sounds huge. What is such a long span?
>
> **Response** When referring to "length," we are specifically addressing the "number of characters" in a post. In an exceptional instance, the longest post, comprising 400 characters, was identified as a "smears" campaign targeting an individual. This post aimed to inflict reputational harm throughout its entirety, leading to its extended length.
>
> ---
>
> **Question-H** Table 3. How is this translated? What is the quality of such translation?
>
> **Response** The process of translating the code-switched text (comprising both English and Roman Urdu) into English involves utilizing Google Cloud Platform's GCP Cloud Translation API. This API is capable of translating text across more than 100 language pairs. It relies on a Google pre-trained Neural Machine Translation (NMT) model and can automatically detect the presence of Roman Urdu words within the English code-switched text. Although the API does not explicitly mention a qualitative measure for translation, such as the Bleu score, we performed a manual assessment of 20 posts, encompassing both short and long sentences. Based on our observations from these evaluations, we found that the model provided reasonably accurate translations. Consequently, we chose to proceed with utilizing its translation services.
>
> ---
>
> **Question-I** "i.e., each example can be assigned zero, one, or multiple labels" What is an example? The whole post?
>
> **Response** We apologize for the confusion. An example is the entire post/text (This post/text/example can have zero, one, or multiple labels)
>
>
> &nbsp;
> &nbsp;
> &nbsp;
> &nbsp;
> &nbsp;
>
> ---
>
> **Typos Grammar Style And Presentation Improvements**
>
> **Response** We express our gratitude to the reviewers for their diligent reading and the considerable effort they invested in their review. We will address all of the issues and revise the manuscript. We will also review the problematic references and rectify the mistakes.

---

### Official Review · Reviewer_ut2p · 2023-08-04

**Soundness:** 2

**Excitement:**

1: Poor: I cannot identify the contributions of this paper, or I believe the claims are not sufficiently backed up by evidence. I would fight to have it rejected.

**Paper Topic And Main Contributions:**

The pape proposes a new dataset for propaganda in Latin-Urdu detection with the particularity of including code-switching words in English.

To sum up it is the common new dataset paper, which describes the way of taking the documents, the annotation and some experiments to validate the new corpus. I have to say that the use of code-switching words is not novel, and it is known that it helps in multilingual scenarios, and in particular in social media genre [1]. For this reason, the multilingual LM are the ones with higher performance.

The differences among using the code-switched words and not using them (translation approach) are not large, when the same learning models (LM) are used. I did not considered the results with different models because they are not comparable. Therefore, in this case, I do not see the contribution of code-switched words. This may be motivated by the low presence of code-switched words. Therefore, I do not see the real contribution of code-switched words for the classification of propaganda. At least, I see that multlingual LM can work with code-switched words, which is known [1].

To sum up,  I do not see this dataset paper in EMNLP, as well as it lacks of some relevant details to assess the contribution of this new dataset for the research community.

References

[1] Camacho-Collados, J., Doval, Y., Martínez-Cámara, E., Espinosa-Anke, L., Barbieri, F., & Schockaert, S. (2020, May). Learning cross-lingual word embeddings from twitter via distant supervision. In Proceedings of the international AAAI conference on web and social media (Vol. 14, pp. 72-82).

**Questions For The Authors:**

- Do the code-switched words contribute to the propaganda classification with monolingual language models or supervised classification models?
- Do you plan to release the corpus?

**Reasons To Accept:**

- The creation of a new dataset in a non-English language.

**Reasons To Reject:**

- It is not motivated the relevance of the new dataset.
- The experimental setup does not show the contribution of code-switched words.
- The unique evidence is that multilingual language models do a good job with code-switched words, which is known by the community.

**Reproducibility:**

2: Would be hard pressed to reproduce the results. The contribution depends on data that are simply not available outside the author's institution or consortium; not enough details are provided.

**Reviewer Confidence:**

4: Quite sure. I tried to check the important points carefully. It's unlikely, though conceivable, that I missed something that should affect my ratings.

---

> ### Author Rebuttal · Authors · 2023-08-29
>
> **Reasons to Reject**
>
> ---
> ---
>
> **Comment** It is not motivated the relevance of the new dataset.
>
> **Response** Significant efforts have been made in investigating the identification of propaganda techniques in high-resource languages like English; however, low-resource languages have not received sufficient attention in this domain. A considerable number of social media users from low-resource languages often engage in code-switching while expressing their opinions online. Given the presence of these users on social media platforms, it is important to have a dataset for effectively studying the identification of propaganda techniques on code-switched text. The introduction of a new code-switched dataset marks a step in this direction.
>
> ---
>
> **Comment** The experimental setup does not show the contribution of code-switched words.
>
> **Response** We agree that we do not show the influence of code-switched words as we did not annotate the words with classes "English" and "Roman Urdu". The data was only annotated with "propaganda techniques" on a fragment-level. Due to this reason, we could not observe the distinct contributions of Roman Urdu words versus English words within our models.
>
> ---
>
> **Comment** The unique evidence is that multilingual language models do a good job with code-switched words, which is known by the community.
>
> **Response** While the community acknowledges the capabilities of multilingual language models in handling code-switched words, it's important to clarify that detecting propaganda techniques using multilingual models on code-switched text is more intricate than a simple comparison with monolingual models on translated content. Multilingual models have limitations, especially with low-resource languages, which can complicate the task of identifying subtle nuances inherent in propaganda, making it a challenging task. Therefore, we give attention to this comparison in our concluding remarks. It's also worth noting that these discussions aren't the primary focus of our highlighted contributions in the paper.
>
>
>
> &nbsp;
> &nbsp;
> &nbsp;
> &nbsp;
> &nbsp;
>
> **Questions for the Author**
>
> ---
> ---
>
> **Question-A** Do the code-switched words contribute to the propaganda classification with monolingual language models or supervised classification models?
>
> **Response** Please note that all of the models used in this work fall under the category of supervised classification (i.e. detecting propaganda techniques in multi-class multi-label task settings on code-switched examples/texts). Secondly, we did not annotate the data (individual words of examples) with classes such as "English" and "Code-switched (Roman-Urdu)", but rather with propaganda techniques. Due to the non-availability of labels of individual words (i.e. "English" and "Code-switched (Roman-Urdu)"), we have not conducted a study to see the impact of only "code-switched" words on the identification of propaganda techniques in monolingual, multilingual and cross-lingual language models.
>
> ---
>
> **Question-B** Do you plan to release the corpus?
>
> **Response** Yes, as mentioned in the abstract, we will make the corpus publicly available once the EMNLP anonymity period concludes. Additionally, we will release the code for our custom-developed annotation platform, used for fragment-level annotation of texts. Furthermore, we will make the code for our models and comprehensive instructions accessible to ensure complete reproducibility.
>
>
> &nbsp;
> &nbsp;
> &nbsp;
> &nbsp;
> &nbsp;
>
> **Reproducibility**
>
> ---
> ---
>
> **Comment** Would be hard pressed to reproduce the results. The contribution depends on data that are simply not available outside the author's institution or consortium; not enough details are provided.
>
> **Response** We assure the reviewer that the reproducibility of our work will be a straightforward process. Once the anonymity period concludes, we publicly release the code and dataset. The dataset will be accessible in both JSON and CSV formats. Furthermore, we will provide the train and test splits that we use along with the seed value. We will also provide comprehensive instructions and Python commands for executing each model for both training and inference. Specific hyperparameters will be presented in the Python commands which are the same as those available in Section 4.3, "Experimental Settings." Moreover, we have included the code for generating propaganda techniques using GPT-3.5 Turbo, complete with the corresponding prompts. The prompt for GPT-3.5 can also be located in the Appendix C.

---

### Official Review · Reviewer_KqAR · 2023-08-11

**Soundness:** 3

**Ethical Concerns:**

Yes

**Excitement:**

4: Strong: This paper deepens the understanding of some phenomenon or lowers the barriers to an existing research direction.

**Justification For Ethical Concerns:**

The paper uses a set of workers to collect social media post and two annotators. However, it lacks discussion how these annotators were selected, pay, and consent. It does not state whether the annotation process passed an institutional ethics board or policy.

_In the rebuttal, the author has promised to include this information in the paper - without having read this it is difficult to say whether it will now meet the ethics policy_

**Paper Topic And Main Contributions:**

Social media users who speak multiple languages may code switch, i.e. post in a mixture of two languages. This paper investigates the detection of propaganda in code-switched English and romanised Urdu.

The main contributions are:
- A dataset propaganda techniques in English/Roman Urdu tweets, annotated at span level via a rigorous process
- Results for training a variety of model configurations (including specific roman Urdu models) in different scenarios (out of domain training, multilingual/monolingual and few-shot prompting)
- Overall findings that translating romanised Urdu is insufficient and the multilingualism should be used directly

**Questions For The Authors:**

A: Why was 20-shot chosen for the GPT model? This seems a high number to use as the only experiment.

B: BERT, a model which should have minimal knowledge of roman Urdu, appears to in some cases outperform models that should have bilingual understanding (e.g. the F1 of model 7 is higher than 9). Was anything done to understand why this happens, does this performance come from solely the English parts of the sentence?

C: The annotation process is rigorous and it says "there were rarely disagreements after the conflict was discussed". Yet, the Krippendorff alpha is still (slightly) below the 0.8 threshold. What caused this, as I would expect if conflicts were discussed and annotations amended, the score would be higher?

D: How will the dataset be distributed in the final version of the paper, given that 60% of the dataset is Twitter who require datasets only contain the ID and end users must retrieve text, which is now difficult given Twitter API changes?

E: Is GCP Cloud Translate capable of actually translating roman Urdu? I tried the first example from Appendix A in Google Translate and it seems unable to - is GCP Cloud Translate better? The limitations discussion for translation only discuss that propaganda techniques may be affected when translated.

---

_Following the rebuttal, I am satisfied with the answers to these questions_

**Reasons To Accept:**

- Annotation methodology is excellent. Previous similar datasets (e.g. semeval 2023 task 3) have suffered from significant annotation noise due to the complexities of the problem, but this methodology would appear to improve on this
- A wide selection of models are trialled, including general-purpose and Roman Urdu-specific.
- It considers and rules out translation for this task, a widespread method used in multilingual analysis
- Much propaganda research is for English or other European languages. This work not only considers an uncommon language for work in this field, but also considers it in code switched form, which widens its usefulness for dealing with social media where code switching is common.
- The presentation of results is clear

**Reasons To Reject:**

- The results section lacks discussion on the majority of the models, almost exclusively discussing the highest scoring (11,12 and 13), so the results for the earlier strategies are not very well covered. - _Given the rebuttal, I now understand why and no longer consider this an issue_
- The out-of-domain dataset chosen (Dimitrov et al.'s meme propaganda dataset) seems an odd choice, given it's only in English. Although I see the value of training with an out-of-domain dataset and an English-only dataset, I think combining them makes it difficult to interpret the results. I appreciate that this is where the label schema used comes from, but I question the dataset itself's usage. _I understand the rebuttal argument that data is scarce, and that this is very possibly the best choice of data to use, but I still think the fact this dataset is both English-only and collected from memes makes it problematic to compare_
- There is no discussion of the ethics of the annotation tasks. No information is given on how they were recruited, pay, whether they gave proper informed consent etc. _The authors have promised to include this information in the next revision_

**Reproducibility:**

4: Could mostly reproduce the results, but there may be some variation because of sample variance or minor variations in their interpretation of the protocol or method.

**Reviewer Confidence:**

4: Quite sure. I tried to check the important points carefully. It's unlikely, though conceivable, that I missed something that should affect my ratings.

**Typos Grammar Style And Presentation Improvements:**

Line 228: Citation should not be in parentheses

Table 4: Missing dividing line between models 12 and 13, as in table 3

---

> ### Author Rebuttal · Authors · 2023-08-29
>
> **Reasons to Reject**
>
> ---
> ---
>
> **Comment** The results section lacks discussion on the majority of the models, almost exclusively discussing the highest scoring (11,12 and 13), so the results for the earlier strategies are not very well covered.
>
> **Response** In our experiments, our primary focus is to present a comparison among the four different fine-tuning strategies depicted in Table 3. The comparison we conduct pertains to the fine-tuning of pretrained models using out-of-domain, translated, and code-switched datasets - our three fine-tuning strategies. These comparisons are meaningful only when carried out using monolingual, multilingual (including bilingual), and cross-lingual language models which we do in models $M_{1}$ to $M_{10}$ using BERT, mBERT, RUBERT and XLM-RoBERTa. Thus, rather than engaging in an in-depth model-to-model comparison for $M_{1}$ to $M_{10}$, we make a comparison between the different fine-tuning strategies. In addition, we introduce the usage of a LLM (GPT-3.5) without fine-tuning as an approach, and its results are analyzed under $M_{13}$. We then delve into a more detailed discussion of models $M_{11}$, $M_{12}$, and $M_{13}$ which achieve a much better performance in comparison to models  $M_{1}$ to $M_{10}$. These models are highlighted due to their superior performance according to our primary evaluation metric (micro-average f1-score).
>
> ---
>
> **Comment** The out-of-domain dataset chosen (Dimitrov et al.'s meme propaganda dataset) seems an odd choice, given it's only in English. Although I see the value of training with an out-of-domain dataset and an English-only dataset, I think combining them makes it difficult to interpret the results. I appreciate that this is where the label schema used comes from, but I Comment the dataset itself's usage.
>
> **Response** Given the scarcity of labeled data in the propaganda detection domain, particularly with the specific 20 propaganda labels that we use, there was an absence of out-of-domain datasets using code-switched languages, including Roman Urdu and English. Therefore, there are no code-switched out-of-domain datasets available for use. Consequently, for any form of out-of-domain fine-tuning comparison, we must rely on the few existing options from which we choose: the meme dataset (Dimitrov et al.'s meme propaganda dataset) which was in English. In our comparison, we contrast this out-of-domain fine-tuning approach with the code-switched fine-tuning approach on our dataset.
>
> ---
>
> **Comment** There is no discussion of the ethics of the annotation tasks. No information is given on how they were recruited, pay, whether they gave proper informed consent etc
>
> **Response** We make sure to uphold the highest ethical standards in all our research endeavors and confirm that all necessary annotations and procedures were meticulously carried out internally to ensure the validity and integrity of the study. The 4 data collectors, selected by the authors based on the quality of their submissions from a pool of 10 candidates, consented to the collection task and were fairly compensated for gathering 2000 code-switched texts (500 each). We will add this information in the ethics section of revised manuscript.
>
>
>
> &nbsp;
> &nbsp;
> &nbsp;
> &nbsp;
> &nbsp;
>
> **Questions for the Authors**
>
> **Question-A** Why was 20-shot chosen for the GPT model? This seems a high number to use as the only experiment.
>
> **Response** Firstly, we use the GPT model, providing a set of examples within the prompt for four distinct few-shot settings (Shown in Appendix C). We then assess performance across these four settings and observe that the 20-shot configuration yields the highest micro-average F1-Score which is our primary evaluation metric, hence one of the factors motivating its selection. Secondly, another rationale behind opting for the 20-shot setting in the GPT model is that the prompt contains 20 examples. This is the only prompt using representations of all our 20 propaganda techniques. As a result, each propaganda technique is covered by at least one example, further underscoring the suitability of the 20-shot setup.
>
> ---
>
> **Question-B** BERT, a model which should have minimal knowledge of roman Urdu, appears to in some cases outperform models that should have bilingual understanding (e.g. the F1 of model 7 is higher than 9). Was anything done to understand why this happens, does this performance come from solely the English parts of the sentence?
>
> **Response**
> Although we did not specifically delve into this aspect. This performance discrepancy could arise from a combination of factors, potentially including the efficacy of BERT in comprehending the English parts of the sentence and its ability to glean insights from the context that extends beyond individual languages. Further analysis could shed light on the nuanced interactions contributing to BERT's performance in future work.
>
> ---
>
> **Question-C** The annotation process is rigorous and it says "there were rarely disagreements after the conflict was discussed". Yet, the Krippendorff alpha is still (slightly) below the 0.8 threshold. What caused this, as I would expect if conflicts were discussed and annotations amended, the score would be higher?
>
> **Response** The reviewer raises a valuable question, and we apologize for any lack of clarity. The computation of Krippendorff's alpha was conducted when both annotators compared labels and spans within the text/example, prior to conflict resolution. Our remark "there were rarely disagreements after the conflict was discussed" refers to the case during conflict resolution where there was a conflict on a certain propaganda label, and both annotators discussed and cross-questioned each other. After this discussion, both annotators would almost always reach a consensus on the propaganda label in question and thus there was no conflict left. Additionally, the relatively lower threshold (the Krippendorff's alpha before conflict resolution) can be attributed to the incorporation of both labels and spans in the calculation. The intricacies of handling spans posed a substantial challenge, significantly impacting the overall score. Notably, if the calculation were solely based on labels, the Krippendorff's alpha would surpass 0.9
>
> ---
>
> **Question-D** How will the dataset be distributed in the final version of the paper, given that 60\% of the dataset is Twitter who require datasets only contain the ID and end users must retrieve text, which is now difficult given Twitter API changes?
>
> **Response** The dataset is now in its finalized version, with 1030 code-switched texts, which might be expanded in future research. To prioritize anonymity, we intentionally abstain from collecting ID/user information associated with the collected text/examples. The data collectors (totaling 4)  manually collected the data/posts/texts from public accounts without using the Twitter API. Therefore, any potential modifications to the Twitter API in the future will not impact our forthcoming research or data gathering procedures. It's also worth noting that the collectors received appropriate compensation for their diligent efforts in manually collecting the code-switched texts.
>
> ---
>
>
> **Question-E**
> Is GCP Cloud Translate capable of actually translating roman Urdu? I tried the first example from Appendix A in Google Translate and it seems unable to - is GCP Cloud Translate better? The limitations discussion for translation only discuss that propaganda techniques may be affected when translated.
>
> **Response** Yes, the GCP Cloud Translate effectively translates Roman Urdu to English, which is why we opted for the paid service. We observed that Google Translate did not seem to give proper translations, leading us to choose the paid GCP Cloud Translate API. The quality of the translation decision could be attributed to the paid service's capability to facilitate the translation of Roman Urdu + English to English. While the quality of translation provided by the GCP Cloud Translate API is not a limitation, we acknowledge that the challenges due to the stylistic nuances inherent in the code-switched language are not transferable to the different languages (In our case English). Furthermore, specific propaganda techniques exhibit language-specific attributes, which can result in the loss of the text's propagandistic essence even with a proficient translation from code-switched to English. Here is an example from our dataset of a translated code-switched text (Roman-Urdu) to English using GCP Cloud Translate API to show the quality of translations (Which seem to be of a high quality):
>
> **Code-switched Text** *"See this group of foreign funded women dancing on the roads, phir kekte hain iss mulk mein zalzale kyun aate hain. Ye Khuda ka azaab nhi hai toh aur kya hai?"*
>
> **English Translation** *"Watch this group of foreign-funded women dance in the streets, then see why earthquakes happen in this country. If this is not God's punishment, what else is?"*
>
>
>
> &nbsp;
> &nbsp;
> &nbsp;
>
>
> **Typos Grammar Style And Presentation Improvements**
>
> ---
> ---
>
> **Comment** Line 228: Citation should not be in parentheses Table 4: Missing dividing line between models 12 and 13, as in table 3
>
> **Response** Thank you for your feedback on the improvements. We will address the two corrections in our final manuscript.

---

### Official Review · Reviewer_s1qS · 2023-08-11

**Soundness:** 4

**Excitement:**

4: Strong: This paper deepens the understanding of some phenomenon or lowers the barriers to an existing research direction.

**Paper Topic And Main Contributions:**

This article proposes the task of detecting propaganda in code-switched text, featuring a low-resourced language. The article presents a corpus of 1030 texts, annotated on fragment level with propaganda techniques, and experiments with different types of models. The paper provides useful outcomes regarding which approach is better.

Contributions:
1) a new task proposed
2) a new English - Roman Urdu corpus, annotated with propaganda techniques at the fragment level is presented
3) useful conclusions about which modelling approach is better, are provided

The article is very clearly written, but some details would make it better:

1.	A reference article defining “code-switching” can be added, when introducing the term in the Introduction.
2.	It would be good to add a very short discussion of what topics the texts are on.

**Reasons To Accept:**

1. New task proposed
2. New dataset proposed, while small, it contains a rare combination
3. Meaningful and useful results
4. Very well written article

**Reasons To Reject:**

none

**Reproducibility:**

3: Could reproduce the results with some difficulty. The settings of parameters are underspecified or subjectively determined; the training/evaluation data are not widely available.

**Reviewer Confidence:**

3: Pretty sure, but there's a chance I missed something. Although I have a good feel for this area in general, I did not carefully check the paper's details, e.g., the math, experimental design, or novelty.

---

> ### Author Rebuttal · Authors · 2023-08-29
>
> **Paper Topic And Main Contributions**
>
> ---
> ---
>
> **Comment** A reference article defining “code-switching” can be added, when introducing the term in the Introduction
>
> **Response** Thank you for your feedback and suggestions. Following your recommendations, we intend to revise our manuscript by incorporating the paper mentioned below. This paper defines the concept of Code-Switching and provides a comprehensive overview of various collected datasets, along with the diverse tasks conducted on these datasets.
>
>
> *Sitaram, Sunayana, et al. "A survey of code-switched speech and language processing."*
>
> ---
>
> **Comment** It would be good to add a very short discussion of what topics the texts are on.
>
> **Response** We will revise the manuscript by including additional content about the topics from which these code-switched texts were derived from. Some of the topics covered in these texts include politics, sports (cricket), women's march (Aurat march), feminism, religious views, and gender equality.
>
>
> &nbsp;
> &nbsp;
> &nbsp;
>
>
> **Reproducibility**
>
> ---
> ---
>
>
> **Comment** Could reproduce the results with some difficulty. The settings of parameters are underspecified or subjectively determined; the training/evaluation data are not widely available.
>
> **Response** We assure the reviewer that the reproducibility of our work will be a straightforward process. Once the anonymity period concludes, we publicly release the code and dataset. The dataset will be accessible in both JSON and CSV formats. Furthermore, we will provide the train and test splits that we use along with the seed value. We will also provide comprehensive instructions and Python commands for executing each model for both training and inference. Specific hyperparameters will be presented in the Python commands which are the same as those available in Section 4.3, "Experimental Settings." Moreover, we have included the code for generating propaganda techniques using GPT-3.5 Turbo, complete with the corresponding prompts. The prompt for GPT-3.5 can also be located in the Appendix C.

---

### Official Review · Reviewer_o5WW · 2023-08-12

**Soundness:** 2

**Excitement:**

3: Ambivalent: It has merits (e.g., it reports state-of-the-art results, the idea is nice), but there are key weaknesses (e.g., it describes incremental work), and it can significantly benefit from another round of revision. However, I won't object to accepting it if my co-reviewers champion it.

**Paper Topic And Main Contributions:**

The authors introduced a novel task - "Detecting propaganda in a Code-switching Scenario", which was not previously explored before. The dataset seems novel and the experimental analysis is well presented. However, the two languages are English and 'Transliterated' Urdu, hence labelling it as a Code-switching phenomena, might not be factually right.

**Questions For The Authors:**

Why does the annotation training work?
What if the dataset includes actual Urdu written in native script, instead of Roman?
How did you pick and choose the 'propaganda related texts' from social media?


**Reasons To Accept:**

Novelty of the task and dataset.

**Reasons To Reject:**

The dataset is very small. Most datasets in the same domain have significantly more amount of data.
The annotator training procedure is a completely new approach and not directly motivated from any previous study. It's unclear why this training would make an annotator eligible enough to annotate the data.
The motivation behind the experimental analysis on the out-of-domain meme dataset is unclear.

**Reproducibility:**

4: Could mostly reproduce the results, but there may be some variation because of sample variance or minor variations in their interpretation of the protocol or method.

**Reviewer Confidence:**

4: Quite sure. I tried to check the important points carefully. It's unlikely, though conceivable, that I missed something that should affect my ratings.

---

> ### Author Rebuttal · Authors · 2023-08-29
>
> **Reasons to Reject**
>
> ---
> ---
>
> **Comment** The dataset is very small. Most datasets in the same domain have significantly more amount of data.
>
> **Response** We acknowledge that the dataset is indeed small, but it is important to emphasize that the emphasis here is on ensuring the highest quality rather than sheer quantity. While a larger dataset might offer more data points, a smaller dataset curated with meticulous attention can provide a more valuable foundation for analysis and learning. By prioritizing quality over quantity, the data collected and included in the dataset has undergone rigorous scrutiny, filtering, and validation processes.
>
> It is important to note that within the specific domain of identifying propaganda techniques at the span-level, the task is inherently challenging. It demands comprehensive training, which our annotators underwent. This training encompassed reviewing over 250 existing texts labeled with the 20 propaganda techniques in English from a previous dataset. Subsequently, the actual annotation process commenced, which required intense focus and diligence for each span of text. This complete endeavor of annotation training and the annotation on our dataset spanned approximately two and a half months, during which two annotators underwent rigorous training and became eligible to annotate. The annotation training task itself extended over a period of two weeks and the the actual annotation process on our dataset took approximately 2 months (two and a half months in total). This duration was influenced by the limited resources dedicated to this meticulous task. There is a scarcity of individuals who are fluent in both Urdu + English and possess the ability to critically analyze and identify propaganda in the text (unless they undergo proper training).
>
> ---
>
> **Comment** The annotator training procedure is a completely new approach and not directly motivated from any previous study. It's unclear why this training would make an annotator eligible enough to annotate the data.
>
> **Response** The annotator training method ensures that both annotators thoroughly review the definitions and examples from a previously labeled dataset. The method involves a training sequence where annotators first work through one set of labeled texts. Following this, they receive feedback from a domain expert (an individual experienced in this task who has annotated texts before). Subsequently, they proceed to train on another set, leveraging the training and feedback from the first set. We found this approach to be remarkably effective. The shared learning experience during the annotation of both sets significantly enhanced the consensus between the annotators. This positive impact on consensus became particularly evident as they trained on the second set.
>
> ---
>
> **Comment** The motivation behind the experimental analysis on the out-of-domain meme dataset is unclear.
>
> **Response** Existing literature shows that training on an out-of-domain dataset can enhance the model's ability to generalize and perform well on different types of data, including one's own dataset. This can help the model capture broader patterns and nuances in propaganda techniques. The comparison between out-of-domain and in-domain performance can provide insights into the model's adaptability towards new datasets. If the model performs well on both datasets, it indicates that it has successfully learned transferable features. Following these observations from existing literature, we used a model fine-tuned on an out-of-domain dataset for inference on our own code-switched dataset. Results indicate that fine-tuning an out-of-domain dataset does not help improve the performance of propaganda technique identification on our code-switched dataset. This can be attributed to the nature of the out-of-domain dataset, which consists of text captions from meme images. Without the accompanying images, the contextual information within the text may degrade which leads to poor performance. We will add motivation behind experimental analysis on an out-of-domain dataset in the manuscript.
>
>
> &nbsp;
> &nbsp;
>
> **Questions for the Authors**
>
> ---
> ---
>
> **Question-A** Why does the annotation training work?
>
> **Response** The annotator training method ensures that both annotators thoroughly review the definitions and examples provided by a previously labeled dataset. Both annotators undergo training using more than 250 labeled texts which include examples for each of the 20 propaganda techniques. These 250 texts are split into two sets where they work through one set of labeled texts and receive feedback from a domain expert (an individual who has previously trained for this task and annotated texts). Following this, they proceed to label another set, benefiting from the training and feedback received for the first set. We have found this method to be highly effective. The collaborative learning experience while annotating both sets significantly enhanced the consensus between the annotators. This became particularly evident as they worked through the second set. Furthermore, during the actual annotation of the dataset, conflicts between the annotators were rare and easily resolved. The Krippendorff's alpha score was notably favorable, especially considering that the annotation involved span-level precision on which Krippendorff's alpha usually doesn't score too well.
>
>
> ---
>
> **Question-B** What if the dataset includes actual Urdu written in native script, instead of Roman?
>
> **Response** In our dataset, we collected only posts/texts/examples written in Roman Urdu + English and did not include those posts/texts/examples that were written in the native Urdu script. However, the annotators were well-versed in native Urdu, Roman Urdu, and English. If needed, annotating native Urdu texts wouldn't be an issue. The reason for the choice of Roman Urdu was its abundance on social media compared to the native script, as many people prefer to type in Latin script rather than Urdu's native script because of its convenience in typing.
>
> ---
>
> **Question-C** How did you pick and choose the 'propaganda related texts' from social media?
>
> **Response** The social media texts were manually hand-picked from social media accounts and public groups and pages on different propagandistic topics, such as politics, cricket, woman’s march (Aurat march), feminism, religious views, gender equality and more whose texts were full of propagandistic posts, examples, and content. Furthermore, the data collectors prioritized posts/texts that displayed a higher degree of richness in propaganda techniques. This approach aimed to encompass a wider array of propaganda techniques within our limited dataset. Nonetheless, we intentionally also included posts/texts/examples devoid of propaganda labels. This inclusion prevents the model from consistently attempting to identify a propaganda technique when it is not inherently present.

---

### Official Review · Reviewer_X7cv · 2023-08-12

**Soundness:** 5

**Excitement:**

4: Strong: This paper deepens the understanding of some phenomenon or lowers the barriers to an existing research direction.

**Missing References:**

Hammad Rizwan, Muhammad Haroon Shakeel, and Asim Karim. 2020. Hate-Speech and Offensive Language Detection in Roman Urdu. In Proceedings of the 2020 Conference on Empirical Methods in Natural Language Processing (EMNLP), pages 2512–2522, Online. Association for Computational Linguistics.

**Paper Topic And Main Contributions:**

This paper studies how propaganda is communicated on social media in a code-switched setting. The authors develop a novel corpus of social media posts which contain code-switched content in Roman Urdu and English and annotate spans of text in each example for many different kinds of propaganda techniques. Finally, they perform experiments to detect this propaganda using a variety of different models from different model families.

**Questions For The Authors:**

A. What instructions were given to the collectors for collecting this data? Did they search for certain keywords or keep an eye on certain online conversations? How did you ensure balance across different topics?
B. Did you analyze the distribution of topics across the dataset and look at which topics are more prone to certain types of propaganda? There is a mix of social and political examples among the ones provided in the appendix and it would be interesting to see which techniques are more prevalent in each.
C. Did you explore normalizing the spelling variations of Roman Urdu words to improve the performance of the classification models since they’re operating on a rather small dataset?
D. Have you considered the issue of people being able to search the examples to trace their origins on social media and what impact that might have on the original posters?

**Reasons To Accept:**

Propaganda on social media is a major problem but little attention has been paid to it in the setting of low-resource languages from the Global South, especially when the way the language is commonly written on the internet differs from its original style and script. This paper presents an important study of the phenomenon of online propaganda in Roman Urdu, a major language in South Asia. The dataset itself is a very useful resource and has been annotated in detail with a good analysis of the different propaganda techniques covered in it. The models trained can prove useful for extracting more examples of propaganda from the internet to better understand how it manifests itself in different socio-political conversations. Finally, the paper is very well-written and easy to understand, and should prove very useful for future researchers interested in studying this topic.

**Reasons To Reject:**

None.

**Reproducibility:**

4: Could mostly reproduce the results, but there may be some variation because of sample variance or minor variations in their interpretation of the protocol or method.

**Reviewer Confidence:**

5: Positive that my evaluation is correct. I read the paper very carefully and I am very familiar with related work.

**Typos Grammar Style And Presentation Improvements:**

A. The aspect ratio of the screenshots in figure B.2 doesn’t match.

---

> ### Author Rebuttal · Authors · 2023-08-29
>
> **Questions for the Authors**
>
> ---
> ---
>
>
> **Question-A:** What instructions were given to the collectors for collecting this data? Did they search for certain keywords or keep an eye on certain online conversations? How did you ensure balance across different topics?
>
> **Response:** We introduced data collectors to propaganda techniques by presenting them with various examples from existing datasets in English *(Dimitrov, Dimitar, et al. "Detecting propaganda techniques in memes.")*, as well as from posts that combined English and Roman Urdu. These code-switched posts/texts were gathered by the authors from social media platforms. The purpose was to guide the collectors on the types of posts they should look for. The data collectors were already proficient in both languages i.e, native/Roman Urdu and English.
>
> Afterwards, the collectors were tasked with finding posts on specific topics that exhibited signs of propaganda. These topics included, but were not limited to, politics, cricket, women's march (Aurat march), feminism, religious views, and gender equality. The data collectors were instructed to collect a roughly equal number of posts from each topic. This approach aimed to prevent potential biases during the fine-tuning of pre-trained models.
>
> Throughout the process, the collected data was available to the authors in real-time. Regular feedback was provided to the data collectors to maintain the quality of the examples/posts. This feedback focused on the presence of propaganda techniques and the percentage of code-switched words, ensuring that the examples/posts met the established standards.
>
> ---
>
> **Question-B:** Did you analyze the distribution of topics across the dataset and look at which topics are more prone to certain types of propaganda? There is a mix of social and political examples among the ones provided in the appendix and it would be interesting to see which techniques are more prevalent in each?
>
> **Response:** We didn't instruct the data collectors to include the topics for the posts or examples during the data collection process. As a result, we lack information about the distribution of topics and propaganda techniques across these topics. However, we have shown the distribution of propaganda techniques in the overall annotated dataset.
>
>  ---
>
> **Question-C:** Did you explore normalizing the spelling variations of Roman Urdu words to improve the performance of the classification models since they’re operating on a rather small dataset?
>
> **Response:** While this technique could enhance the quality of embeddings for most Roman Urdu words and minimize noise and inconsistencies in our text data, we made the decision not to proceed with its implementation. This is because real-world textual data often contains errors, typos, and irregularities, particularly these inconsistencies are more pronounced for languages written in non-native scripts like Roman Urdu.
> To ensure our models remain resilient to variations in spelling, we chose not to alter any spellings within the text. However, in the interest of model cleanliness, we did remove unnecessary punctuation, which tends to be overused in social media posts.
> Furthermore, we did not label the words as "English" or "Roman Urdu". Our data was solely annotated with "propaganda technique" on a fragment-level basis. Consequently, we were unable to assess the impact of "spelling variations of Roman Urdu words" on model performance.
>
>  ---
>
>
> **Question-D:** Have you considered the issue of people being able to search the examples to trace their origins on social media and what impact that might have on the original posters?
>
> **Response:** While there's a chance that the source user could be identified by searching for the post or example on social media, it's important to note that we have taken precautions to ensure privacy. We intentionally refrained from gathering source links, user IDs/names, and incorporating them into our dataset. We have also withheld information about which social media platform each post/example originates from.
> Furthermore, our data collection focused on publicly accessible posts that fall within the scope of "Public Information," as defined by the data policies of platforms (Twitter, Facebook, Instagram and YouTube). Users of these social media platforms have consented to their terms and conditions regarding the information they publicly share. It's worth highlighting that we did not collect data from any non-public groups or sources.
>
>
>  ---
>
>
> **Typos Grammar Style And Presentation Improvements**
>
> **Comment** The aspect ratio of the screenshots in figure B.2 doesn’t match
>
> **Response** We appreciate your suggestion and will work on enhancing the visual presentation of the figure to achieve a more professional look in the revised manuscript. Thank you for your feedback.
>
> &nbsp;
>
> **Missing Reference Response** We will ensure that we include the following reference in our revised manuscript, as it is indeed pertinent to the work we have conducted. Thank you for your suggestion.
>
> *Hammad Rizwan, Muhammad Haroon Shakeel, and Asim Karim. 2020. Hate-Speech and Offensive Language Detection in Roman Urdu. In Proceedings of the 2020 Conference on Empirical Methods in Natural Language Processing (EMNLP), pages 2512–2522, Online. Association for Computational Linguistics*

---

### Meta-Review · Area_Chair_2RE6 · 2023-09-18

**Recommendation:** 4

**Metareview:**

The reviews for this paper contain rather divided opinions, some arguing the importance of this contribution, others that it is insufficient.
Reviewers X7cv and  s1qS praise the paper for addressing a significant problem, offering a well-annotated dataset, and presenting a clear and accessible write-up. They find no reasons to reject the paper. I tend to agree with the importance of the proposed resource.
Reviewer KqAR had concerns about discussion of certain choices as well as ethics concerns, however most of their questions were answered through the rebuttal and turned to a positive outlook.
Overall, authors provided a thorough and convincing  responses in the rebuttal period to all of the reviewers.
Reviewer YUyd provided important clarification that while the dataset is novel and challenging, the task of detecting propaganda itself is existing, and finds the contribution mediocre.  Reviewer o5WW is concerned with the small size of the dataset and explanations behind some experimental choices.
Overall, I would recommend the authors to include the reviewers suggestions, answers to their questions, clarifications into the text of the paper.

---

### Decision · Program_Chairs · 2023-10-07

**Decision:**

Accept-Main

**Comment:**

The reviews for this paper contain rather divided opinions, some arguing the importance of this contribution, others that it is insufficient.
Reviewers X7cv and  s1qS praise the paper for addressing a significant problem, offering a well-annotated dataset, and presenting a clear and accessible write-up. They find no reasons to reject the paper. I tend to agree with the importance of the proposed resource.
Reviewer KqAR had concerns about discussion of certain choices as well as ethics concerns, however most of their questions were answered through the rebuttal and turned to a positive outlook.
Overall, authors provided a thorough and convincing  responses in the rebuttal period to all of the reviewers.
Reviewer YUyd provided important clarification that while the dataset is novel and challenging, the task of detecting propaganda itself is existing, and finds the contribution mediocre.  Reviewer o5WW is concerned with the small size of the dataset and explanations behind some experimental choices.
Overall, I would recommend the authors to include the reviewers suggestions, answers to their questions, clarifications into the text of the paper.